# Structural basis for processive daughter-strand synthesis and proofreading by the human leading-strand DNA polymerase Pol ε

Johann J. Roske ● & Joseph T. P. Yeeles ● ✉

During chromosome replication, the nascent leading strand is synthesized by DNA polymerase epsilon (Pol ε), which associates with the sliding clamp processivity factor proliferating cell nuclear antigen (PCNA) to form a processive holoenzyme. For high-fidelity DNA synthesis, Pol ε relies on nucleotide selectivity and its proofreading ability to detect and excise a misincorporated nucleotide. Here, we present cryo-electron microscopy (cryo-EM) structures of human Pol ε in complex with PCNA, DNA and an incoming nucleotide, revealing how Pol ε associates with PCNA through its PCNA-interacting peptide box and additional unique features of its catalytic domain. Furthermore, by solving a series of cryo-EM structures of Pol ε at a mismatch-containing DNA, we elucidate how Pol ε senses and edits a misincorporated nucleotide. Our structures delineate steps along an intramolecular switching mechanism between polymerase and exonuclease activities, providing the basis for a proofreading mechanism in B-family replicative polymerases.

During eukaryotic DNA replication, the replicative helicase CMG (CDC45–MCM–GINS) unwinds the parental DNA double helix to generate single-stranded DNA templates for daughter-strand synthesis. The majority of DNA synthesis is catalyzed by three polymerases (Pols) of the B-family. Pol α–primase generates short primers that are subsequently extended by Pol δ and Pol ε. Pol δ replicates the lagging strand while Pol ε continuously extends the nascent leading strand[1–7].

For processive synthesis, Pol δ and Pol ε depend on the proliferating cell nuclear antigen (PCNA) sliding clamp, a homotrimeric ring-shaped protein that encircles and slides along the nascent DNA double helix[8–12]. Pol δ binds PCNA through a PCNA-interacting peptide (PIP) box motif in the large Pol subunit p125 (or POLD1)[13,14]. The catalytic subunit of human Pol ε, POLE1, also contains a conserved PIP box motif that is predicted to mediate the interaction with PCNA[15,16]. Yet, while crystal structures of the Pol ε catalytic domain (Pol ε cat) from *Saccharomyces cerevisiae* revealed the architecture and mode of engagement with DNA and an incoming nucleotide[17,18], no structural

information on human Pol ε cat is available to date and the interaction between Pol ε and PCNA has not been visualized for any species.

At the replication fork, PCNA also acts as a recruitment platform for various factors that primarily interact through their PIP box motifs that bind to hydrophobic and Q pockets on any of the PCNA protomers[19–21]. For Okazaki fragment maturation at the nascent lagging strand, the Pol δ–PCNA assembly can recruit 'toolbelt' factors such as FEN1 to an unoccupied protomer of the trimeric PCNA ring[13]. While it was demonstrated that no such functional interaction exists between FEN1 and the Pol ε–PCNA holoenzyme[22], it remains unclear how PIP-box-containing toolbelt factors are prevented from interfering with continuous leading-strand synthesis.

High-fidelity DNA Pols exert strict selectivity toward the incoming nucleotide for prechemistry quality control and are additionally equipped with proofreading capability (3′–5′ exonuclease activity) to excise incorrectly inserted nucleotides[23]. The polymerase active site (*pol* site) of Pol ε harbors 'steric gate' and 'sensor' features that

MRC Laboratory of Molecular Biology, Cambridge, UK. ✉e-mail: jyeeles@mrc-lmb.cam.ac.uk

discriminate against ribonucleotides[24–26]. Studies of related Pols have shown how these features work in conjunction with kinetic checkpoints within a conformational change of the Finger domain to actively shape the *pol* site and ensure correct geometry and base pairing of the incoming nucleotide before phosphoryl transfer[27–29]. The kinetics of postinsertion quality control have been measured for human Pol α, which displays a slower catalysis rate and reduced affinity for the incoming nucleotide when bound to a DNA substrate with a mismatch in the postinsertion site[30]. Similarly, the inefficiency with which exonuclease-deficient Pol ε extends a nascent strand that contains a mismatch at the 3′ end is consistent with additional fidelity checkpoints after insertion[23,31].

For proofreading and mismatch excision, the 3′ end of the nascent strand must be partially melted from the template strand and guided toward the Pol ε exonuclease domain active site (*exo* site), which is located almost 40 Å away from the *pol* site. Biochemical studies of *S. cerevisiae* Pol ε identified residues important for activity switching located in the Thumb domain[31] but the molecular mechanisms underlying the detection of an incorrect nucleobase after insertion and its subsequent transfer between the active sites remain elusive. Recent studies of mitochondrial Pol γ, of the A-family, identified intermediate conformers along the transition between polymerization and editing activities and proposed mechanisms for proofreading[32,33]. However, the different architecture of A-family DNA Pols locates the *exo* site on the opposite side of the *pol* site compared with B-family DNA Pols[23], implying different underlying mechanisms for activity switching.

To address the open questions outlined above, we solved cryo-electron microscopy (cryo-EM) structures of human Pol ε bound to PCNA, DNA and an incoming nucleotide. Pol ε cat binds PCNA through extensive interactions through its PIP box and forms additional contacts with the remaining two PCNA protomers through the P domain and an insertion in the Thumb domain. This tripartite interaction involves features that are unique to Pol ε and gives rise to a holoenzyme with an architecture that appears to be specialized for leading-strand synthesis. To investigate the structural mechanism underlying error detection and proofreading, we determined a series of cryo-EM structures of Pol ε cat bound to a mismatch-containing DNA. Our structures reveal Pol ε in distinct states along an intramolecular switching pathway between polymerase and exonuclease activities that provide the basis for a proofreading mechanism in a B-family replicative Pol.

## Results

### Cryo-EM structure of the replicating Pol ε–PCNA holoenzyme
We reconstituted a complex of human Pol ε (exonuclease inactive) and PCNA on a 23-nt nascent strand 38-nt template strand DNA in the presence of 2′,3′-dideoxyadenosine triphosphate (ddATP) and determined three structures by single-particle cryo-EM ranging from 3.6 Å to 3.8 Å in global resolution (Table 1 and Extended Data Fig. 1a–j). The structures primarily differ in the conformation of the Finger domain, which adopts open, closed and 'ajar' conformations that are discussed in detail in the next section. The cryo-EM density maps allowed the unambiguous fit of an atomic model generated from an AlphaFold-Multimer prediction of Pol ε cat (POLE1 residues 1–1250) and three protomer copies of human PCNA[34] and enabled us to model the DNA together with an incoming ddATP in the *pol* site (Fig. 1a,b). The refined model of Pol ε cat encompasses POLE1 residues 27–1197 with only one chain break between residues 182 and 212. The cryo-EM density does not include the noncatalytic domain of Pol ε (POLE1 residues 1200–2286 and subunits POLE2, POLE3 and POLE4), which indicates high flexibility with respect to the catalytic domain, consistent with previous reports[35,36]. However, we observe the noncatalytic domain in distinct two-dimensional (2D) class averages generated from our cryo-EM data, as well as in electron micrographs of a negatively stained sample (Extended Data Fig. 1k–m). Our atomic model of human Pol ε cat closely resembles the crystal structure of the *S. cerevisiae* homolog[17], with an identical domain organization in the

'right-hand'-shaped B-family DNA Pol fold consisting of an N-terminal domain (NTD) and the Palm, Exonuclease (Exo), Finger and Thumb domains[37] (Fig. 1b). Like its budding yeast homolog, human Pol ε cat possesses a unique insertion between the Palm and Finger domains that contacts the nascent DNA; this insertion was found to be important for PCNA-independent processivity and, therefore, termed the processivity domain (P domain)[17]. Density corresponding to an iron–sulfur cluster coordinated by the CysX motif is located at the base of the P domain (Extended Data Fig. 2a)[38,39].

The bound DNA shows three unpaired template nucleobases downstream of the 3′ junction, one base forming a Watson–Crick base pair with the incoming nucleotide in the *pol* site (Extended Data Fig. 2b) and a 24-bp nascent double helix in B-form protruding away from Pol ε and toward PCNA. One dideoxyadenosine was incorporated into the nascent strand and is located at the postinsertion site, acting as the chain terminator.

PCNA adopts the typical trimeric closed ring shape that resembles its previously reported structure[20]. It encircles the double helix of the nascent DNA exiting Pol ε cat, placing its protomers underneath the Palm, Thumb and P domains (Fig. 1). The Pol ε PIP box, which resides immediately C-terminal of the catalytic lobe that ends in the Thumb domain, folds away from the Thumb and contacts the PCNA protomer located underneath the Palm domain (Fig. 1b, left). The conserved Q1180 and the ensuing hydrophobic residues, which fold into a $3_{10}$-helix, dock into the Q and hydrophobic pockets on PCNA, respectively (Fig. 1c), similar to reported PIP box interactions on PCNA[20,21].

In addition to the PIP box, Pol ε cat contacts both remaining PCNA protomers (Fig. 1c,d). A 20-residue insertion that is unique to the Pol ε Thumb domain[17] (residues 1102–1122 in human; Extended Data Fig. 2c) contacts the Q pocket of the second PCNA protomer (Fig. 1d, left) and two α-helices of the P domain are positioned above the Q pocket of the third PCNA protomer (Fig. 1d, right). The ending turn of the second large α-helix of the P domain is wedged between PCNA R210 and Y211, enabled by R210 adopting a rotamer that is different to R210 on the other two PCNA protomers (Extended Data Fig. 2d). The contacts formed with PCNA by the Thumb and P domains are small (~270 Å² and ~670 Å², respectively), mainly mediated by hydrogen bonds and not characteristic of stable protein–protein interactions, potentially allowing respective movement of the contact components. Nevertheless, their locations above the pockets on PCNA establish steric barriers that are incompatible with the engagement of other proteins containing canonical PIP boxes. In comparison, Pol δ only forms interactions with the PCNA protomer underneath the Thumb domain, leaving the remaining PCNA protomers accessible for the recruitment of toolbelt factors such as FEN1 (Extended Data Fig. 2e–g)[13]. This also allows PCNA to tilt around its interaction site with Pol δ (ref. 13), whereas we observe no flexibility or conformational heterogeneity of PCNA with respect to Pol ε cat across our dataset (Extended Data Fig. 1).

### Finger domain closing and nucleoside triphosphate flip
In the previously reported structures of Pol ε bound to DNA and an incoming nucleotide, the Fingers adopt a closed conformation, while the Pol ε apo structure in the absence of DNA shows the Fingers in an open state[17,18,35]. The conformational change of the Fingers shapes the deoxynucleotide triphosphate (dNTP)-binding site and serves as a checkpoint mechanism to ensure correct geometry and base pairing of the incoming nucleotide[27–29]. Single-molecule fluorescence resonance energy transfer studies on A-family Pol I, which shares the core Klenow fold with B-family Pols, showed that the binary complex of Pol and DNA preferentially adopts the open conformation, whereas the binding of a complementary incoming nucleotide induces a conformational change and shifts the equilibrium toward the closed state[40–44]. In the presence of a ribonucleotide or noncomplementary dNTP, Pol I was observed in a semiclosed 'ajar' conformation, which represents a kinetic checkpoint in nucleotide selectivity[41,42,45].

**Table 1 | Statistics of cryo-EM data collection, refinement and validation**

| | Pol ε–PCNA on matched DNA, no detergent | Pol ε–PCNA on matched DNA, 8 mM CHAPSO | | Pol ε–PCNA on mismatched DNA, no detergent | | Pol ε–PCNA on mismatched DNA, 8 mM CHAPSO | |
|---|---|---|---|---|---|---|---|
| **Data collection** | | | | | | | |
| Microscope | FEI Titan Krios | FEI Titan Krios | | FEI Titan Krios | | FEI Titan Krios | |
| Voltage (kV) | 300 | 300 | | 300 | | 300 | |
| Camera | Gatan K3 | Falcon 4i | | Gatan K3 | | Gatan K3 | |
| Magnification | ×105,000 | ×96,000 | | ×105,000 | | ×105,000 | |
| Electron exposure (e⁻ per Å²) | 40.08 | 40.18 | | 36.4 | | 39.9 | |
| Defocus range (µm) | −0.8 to −3.0 | −0.8 to −3.0 | | −1.2 to −3.0 | | −1.2 to −3.0 | |
| Pixel size (Å) | 0.73 | 0.824 | | 0.73 | | 0.73 | |
| Micrographs collected | 13,522 | 12,784 | | 9,659 | | 13,159 | |
| Total extracted particles (no.) | 2.61 million | 3.89 million | | 3.88 million | | 3.52 million | |
| **Reconstruction** | Pol ε–PCNA, Open conf. | Pol ε–PCNA, Ajar conf. | Pol ε–PCNA, Closed conf. | Proofreading Post-Insertion | Proofreading Pol Arrest | Proofreading Frayed Substrate | Proofreading Mismatch Excision |
| EM Data Bank | EMD-50222 | EMD-50223 | EMD-50224 | EMD-50225 | EMD-50226 | EMD-50227 | EMD-50228 |
| PDB | 9F6D | 9F6E | 9F6F | 9F6I | 9F6J | 9F6K | 9F6L |
| Symmetry imposed | None | None | None | None | None | None | None |
| Final particle images (no.) | 175,614 | 69,268 | 92,356 | 119,518 | 44,918 | 43,921 | 18,858 |
| Map resolution$^a$ (Å) 0.143 FSC threshold | 3.6 (3.4) | 3.7 (3.7) | 3.8 (3.7) | 3.3 | 3.9 | 4.2 | 3.9 |
| Map resolution range (Å) 0.5 FSC threshold | 3.0–5.8 | 3.2–6.2 | 3.2–6.1 | 3.3–5.3 | 3.3–9.2 | 3.4–9.5 | 3.4–11 |
| Map sharpening $B$ factor$^a$ (Å²) | −127.2 (−109.7) | −110.3 (−98.2) | −119.3 (−105.8) | −122.1 | −115 | −140 | −93 |
| **Model refinement** | | | | | | | |
| Initial model used (PDB code) | 4M8O 1AXC | 4M8O 1AXC | 4M8O 1AXC | 4M8O | 4M8O 6WJV | 4M8O 6WJV | 4M8O 1CLQ |
| Model resolution (Å) 0.5 FSC threshold | 3.8 | 4.1 | 4.1 | 3.5 | 4 | 4.3 | 4.1 |
| Model composition nonhydrogen atoms | 16,377 | 16,376 | 16,376 | 9,781 | 9,751 | 9,637 | 9,639 |
| Protein residues | 1,918 | 1,918 | 1,918 | 1,118 | 1,118 | 1,118 | 1,118 |
| Nucleotide | 52 | 52 | 52 | 32 | 32 | 26 | 26 |
| Ligands | 1 DDS, 1 SF4, 1 MG | 1 DDS, 1 SF4 | 1 DDS, 1 SF4 | 1 DDS, 1 SF4, 1 CA | 1 SF4 | 1 SF4 | 1 SF4, 2 CA |
| $B$ factors (Å²) | | | | | | | |
| Protein | 40.34 | 91.16 | 98.82 | 27.13 | 86.58 | 125.9 | 111.17 |
| Nucleotide | 108.8 | 155.18 | 160.44 | 65.53 | 149.8 | 198.21 | 162.04 |
| Ligand | 26.52 | 70.08 | 79.11 | 23.21 | 50.08 | 125.48 | 100 |
| R.m.s.d. | | | | | | | |
| Bond lengths (Å) | 0.003 | 0.003 | 0.004 | 0.003 | 0.003 | 0.003 | 0.003 |
| Bond angles (°) | 0.691 | 0.666 | 0.742 | 0.686 | 0.641 | 0.656 | 0.666 |
| Validation | | | | | | | |
| MolProbity score | 0.88 | 0.87 | 1.05 | 0.84 | 0.94 | 1.01 | 0.89 |
| Clashscore | 1.46 | 1.4 | 2.67 | 1.2 | 1.83 | 2.22 | 1.48 |
| Poor rotamers (%) | 0 | 0 | 0 | 0 | 0 | 0.1 | 0 |
| Ramachandran plot | | | | | | | |
| Favored (%) | 99.32 | 99.06 | 99.27 | 98.92 | 98.92 | 97.94 | 98.11 |
| Allowed (%) | 0.68 | 0.94 | 0.73 | 1.08 | 1.08 | 2.06 | 1.89 |
| Disallowed (%) | 0 | 0 | 0 | 0 | 0 | 0 | 0 |

$^a$Statistics for local refinements with mask around Pol ε cat are given in parentheses. DDS, ddATP; SF4, iron–sulfur cluster; CA, Ca²⁺; MG, Mg²⁺.

Strikingly, we observe the Pol ε Fingers in open, closed and ajar conformations, despite the presence of clear density for a matched incoming nucleotide and DNA in all three structures (Fig. 2a,b, Table 1 and Extended Data Fig. 1). Between the open and closed states, the Fingers describe a ~28° rotation (~18° between open and ajar) with the tip of the Fingers traversing ~19 Å distance, while the overall fold of

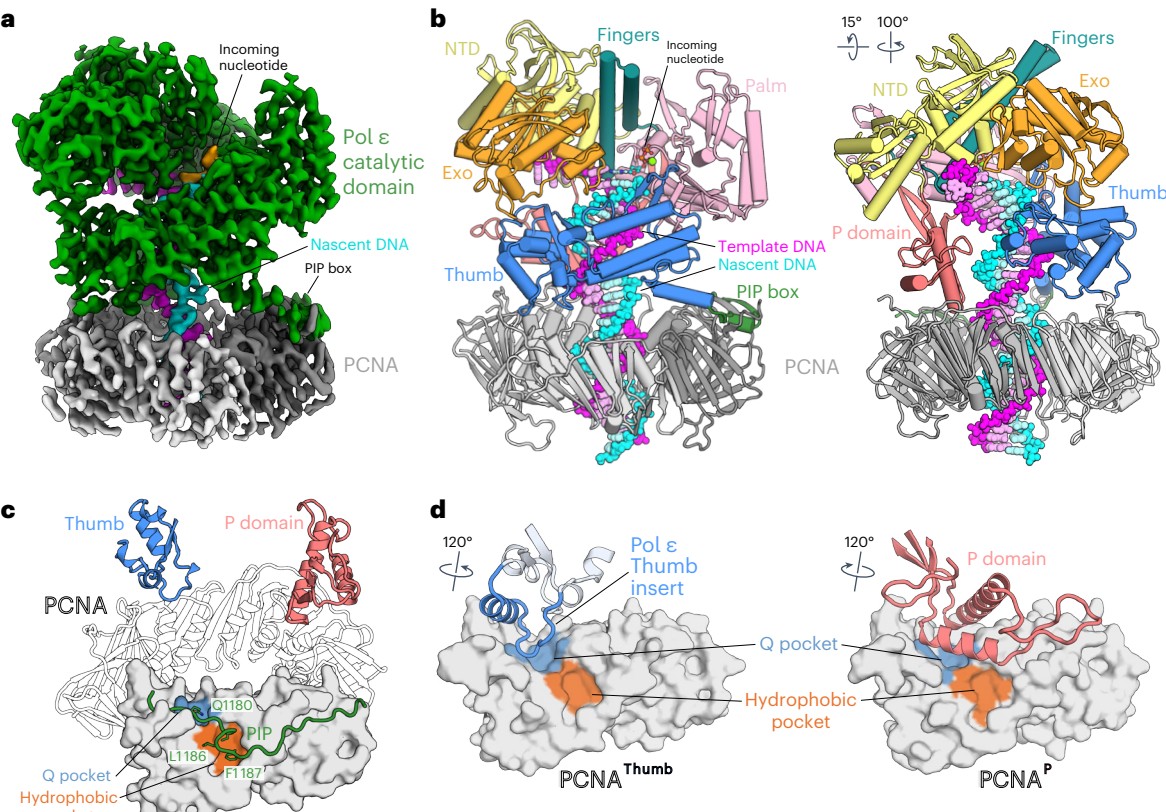

**Fig. 1 | Cryo-EM structure of Pol ε cat bound to DNA and PCNA. a**, Cryo-EM density map of the human Pol ε–DNA–PCNA complex in the open Finger conformation. **b**, Two views of a ribbon model of Pol ε cat in complex with PCNA, substrate DNA and an incoming nucleotide in the open Finger conformation. In this and subsequent figures, unless otherwise stated, domains are colored as follows: NTD, yellow; Exo domain, orange; Finger domain, teal; Palm domain, light pink; P domain, salmon; Thumb domain, blue; PIP box, dark green. DNA template and nascent strands are shown as magenta and cyan spheres, respectively. **c**, Contact sites between PCNA and Pol ε are formed by the PIP box motif and Pol ε unique Thumb insert and P domain. The PCNA protomer that interacts with the PIP box is shown in the front in surface representation. Conserved interface residues of the PIP box are shown as sticks and colored by atom type: carbon, as the respective domain or region; nitrogen, blue; oxygen, red. **d**, Contacts formed by the Thumb insert (blue, left) and P domain (salmon, right) with the adjacent PCNA protomers (PCNA$^{Thumb}$ and PCNA$^P$, respectively). Q and hydrophobic pockets of the PCNA protomers (shown in surface representation) are indicated. Rotation symbols in this figure indicate the view relative to **c**.

Pol ε cat remains unaltered (root-mean-square deviation (r.m.s.d.) of 0.95 Å for 1,076 pairs of Cα atoms, omitting the Finger domain). The Finger-closing motion is hinged at the base of the two α-helices that lies wedged behind the *pol* site between the NTD and Palm domain.

Within the *pol* site, Finger closing correlates with a flip in the triphosphate moiety of the incoming nucleotide (Fig. 2c,d). In the open state, the 5′-triphosphate adopts an 'N-shaped' conformation in which the α phosphate faces away from the 3′ end of the nascent strand and is not positioned for phosphoryl transfer[46]. The α and γ phosphates of the incoming ddATP are coordinated at the Palm domain through the backbone amides of M630 and A629, respectively. Of the Finger domain, only R765 contacts the incoming nucleotide through a salt bridge with the γ phosphate, while residues K769, K809 and sensor N813 remain distant, allowing dissociation and exchange of the nucleotide. In the closed Fingers conformation, the incoming ddATP takes the canonical 'chair-like' conformation in which the β phosphate is coordinated at the M630 amide, while the α phosphate faces toward the 3′ end of the nascent strand, primed for nucleophilic attack (Fig. 2c,d). Additionally, Finger closing positions sensor N813 to hold the nucleobase in place and brings residues K769 and K809 closer to stabilize the triphosphate (Fig. 2c, right).

The ajar conformation illustrates how Finger closing may induce the flip of the triphosphate from the N-shaped to the chair-like conformation. Despite side-chain resolution in the surrounding active site, the triphosphate shows more ambiguous cryo-EM density, suggesting increased heterogeneity and a lack of clear coordination in the ajar

state (Fig. 2d, middle). The density is distinct from the peptide backbone amide of M630, indicating partial release of the α phosphate's coordination at the Palm, which could be promoted by a contact that sensor N813 forms with the α phosphate (Fig. 2c, middle). R765 and K769 moving closer to contact the γ phosphate may additionally aid the release and enable the β phosphate to become coordinated at the Palm with the transition to the closed state.

We note that the use of a 2′,3′-dideoxy nucleotide may be the reason for the triphosphate adopting the N-shaped conformation when the Fingers are open, which was proposed to be the preferred shape of nucleotides missing the 3′-OH and nucleotide analogs with L-stereochemistry[46]. The requirement for the N-shaped triphosphate to release its coordination and rearrange may pose an energetic barrier to the Finger-closing conformational change and thereby enable the visualization of the three distinct conformations in our cryo-EM specimen. Nevertheless, our data show that Pol ε can adopt these conformations in the ternary complex and demonstrate the existence of an ajar transition state in eukaryotic B-family Pols. Additionally, our structures illustrate how the Finger closing motion shapes the *pol* site to potentially manipulate the conformation of the incoming nucleotide triphosphate to position it optimally for phosphoryl transfer.

## Structures of Pol ε activity switching during proofreading

To investigate how Pol ε detects and proofreads a misincorporated nucleotide, we repeated the reconstitution and structure elucidation

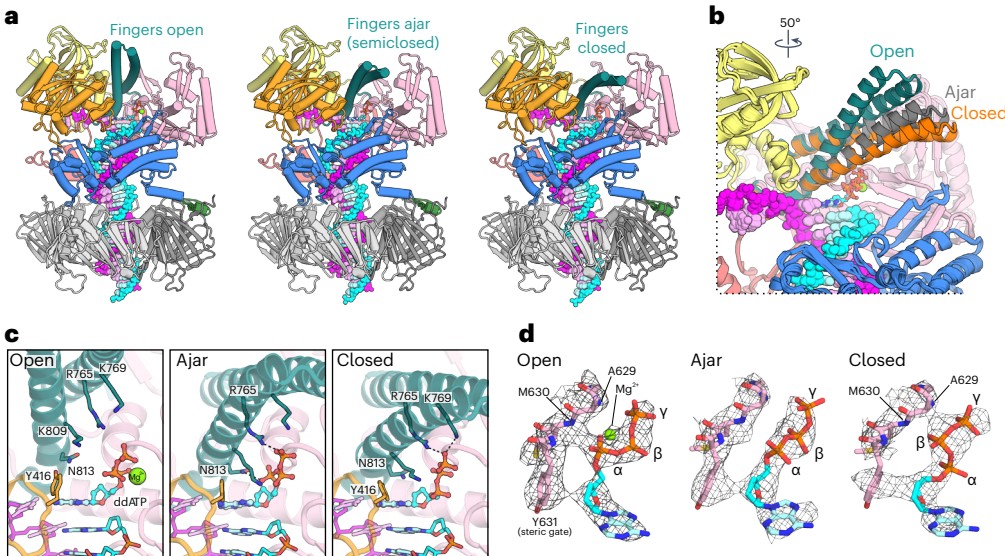

**Fig. 2 | Pol ε Fingers adopt open, ajar and closed conformations. a**, Ribbon models of Pol ε cat in complex with PCNA, substrate DNA and an incoming nucleotide in open, ajar (semiclosed) and closed conformations. **b**, Superposition of the Pol ε cat ternary complex in the three different states shown in **a**. The two α-helices of the Finger domain are indicated for the open (teal), ajar (gray) and closed (orange) conformations. The rotation symbol in this figure indicates the view relative to **a**. **c**, Coordination of the incoming nucleotide within the active site in the respective states. Side chains of the indicated amino acid positions are shown as sticks. Dashed lines indicate electrostatic and hydrogen-bond interactions between the Pol ε Fingers and the triphosphate moiety of the incoming nucleotide. **d**, Cryo-EM density (shown as mesh) for the incoming nucleotide and the coordinating peptide backbone at the surface of the Palm domain.

using DNA that contained a T−C mismatch at the 3′ end of the nascent strand (Fig. 3a). The cryo-EM data gave rise to density maps of Pol ε in four distinct states, representing a Post-Insertion state with the T−C mismatch in the postinsertion site, a Mismatch Excision state with the erroneous base in the *exo* site and two noncatalytic intermediate conformers (Fig. 3, Extended Data Fig. 3 and Table 1).

The four states differ primarily in the conformation of the Thumb domain and the position of the DNA, which is coordinated at either the Thumb or P domain (Fig. 3b–e). All four states also differ in the conformation of the nascent strand–template strand junction, particularly the 3′ end of the nascent strand (Fig. 3f), which we used as the basis to arrange the intermediate states into a plausible order of events in the transition between polymerization and excision activities. In the first intermediate, which we term the Arrest state, the DNA is retracted from the *pol* site but the base pairing remains intact and still resembles the DNA in the Post-Insertion state (Fig. 3c). The second intermediate (the Frayed Substrate state) displays a frayed nascent strand–template strand junction that is more reminiscent of the Mismatch Excision conformer, with the two terminal bases of the nascent strand unpaired from the template. The next sections describe the individual conformers in detail and infer a mechanism for Pol ε proofreading, starting with the detection of the mismatched base after incorporation (Post-Insertion state), followed by its expulsion from the *pol* site (Arrest state), the melting of terminal bases from the template strand (Frayed Substrate state) and the transport of the liberated 3′ end into *exo* site (Mismatch Excision state).

## Pol ε with a T−C mismatch in the postinsertion site

The Post-Insertion state shows Pol ε engaged at the DNA duplex with the terminal T−C mismatch located in the postinsertion site (Fig. 3, 'Post-Insertion'; Fig. 4a). The *pol* site harbors an incoming ddATP that forms a Watson−Crick base pair with the templating thymidine (Fig. 4b). Pol ε cat closely resembles the structure at a matched substrate (r.m.s.d. = 0.87 Å for 1,116 Cα atoms). Notably, we observe Pol ε only in the open conformation and extensive three-dimensional (3D) classification did not reveal any heterogeneity within the Finger domain. The absence of closed or partially closed states, despite the presence of a complementary nucleotide bound in the active site, suggests that the open conformation is favored when Pol ε is bound to a mismatch-containing substrate and may serve as a fidelity checkpoint after mismatch incorporation.

The backbone of the nascent strand displays a subtle distortion at the T−C mismatch in the postinsertion site (Fig. 4c). The terminal cytosine base in the nascent strand is retracted from the templating thymidine and rotated toward the major groove. This increases the gap between the nucleobase faces (Fig. 4d) and renders the base stacking with the incoming nucleotide imperfect. It also increases the distance between respective C1′ from 10.3 Å of the canonical W−C base pair that we observe for Pol ε on the matched substrate to ~11 Å in the T−C mismatch (Fig. 4d). The distortion of the mismatched terminus appears to be translated into the surrounding environment, which shows less well-defined side-chain density despite the slightly higher resolution than in the structure with matched DNA, indicating increased flexibility in this area (Fig. 4e). Most notable are the release of DNA coordination by R955 and of the interactions that Palm residues E858 and D860 form with Thumb residue K954 (Fig. 4e). This rearrangement destabilizes the domain interface between Palm and Thumb, as well as the coordination of DNA near the *pol* site, which may serve as a trigger for subsequent conformational changes in the proofreading mechanism. Consistently, exchange of K954 to alanine destabilizes the nascent 3′ end in budding yeast Pol ε (ref. 31) and compromising the described domain interface at the position corresponding to E858 impairs polymerase activity in bacteriophage RB69 Pol (Extended Data Fig. 4)[47]. A functionally analogous 'fidelity switch' feature was described for A-family polymerase Pol γ, where a side-chain interaction near the Finger joints in the Palm domain is released during proofreading conformational changes[32].

## DNA removal from the *pol* site arrests mismatch extension

In the Arrest state (Fig. 3), the DNA substrate is displaced from the *pol* site. The Thumb domain opens the hand architecture in an outward rotation of ~6 Å (Fig. 5a) and adopts a position that resembles

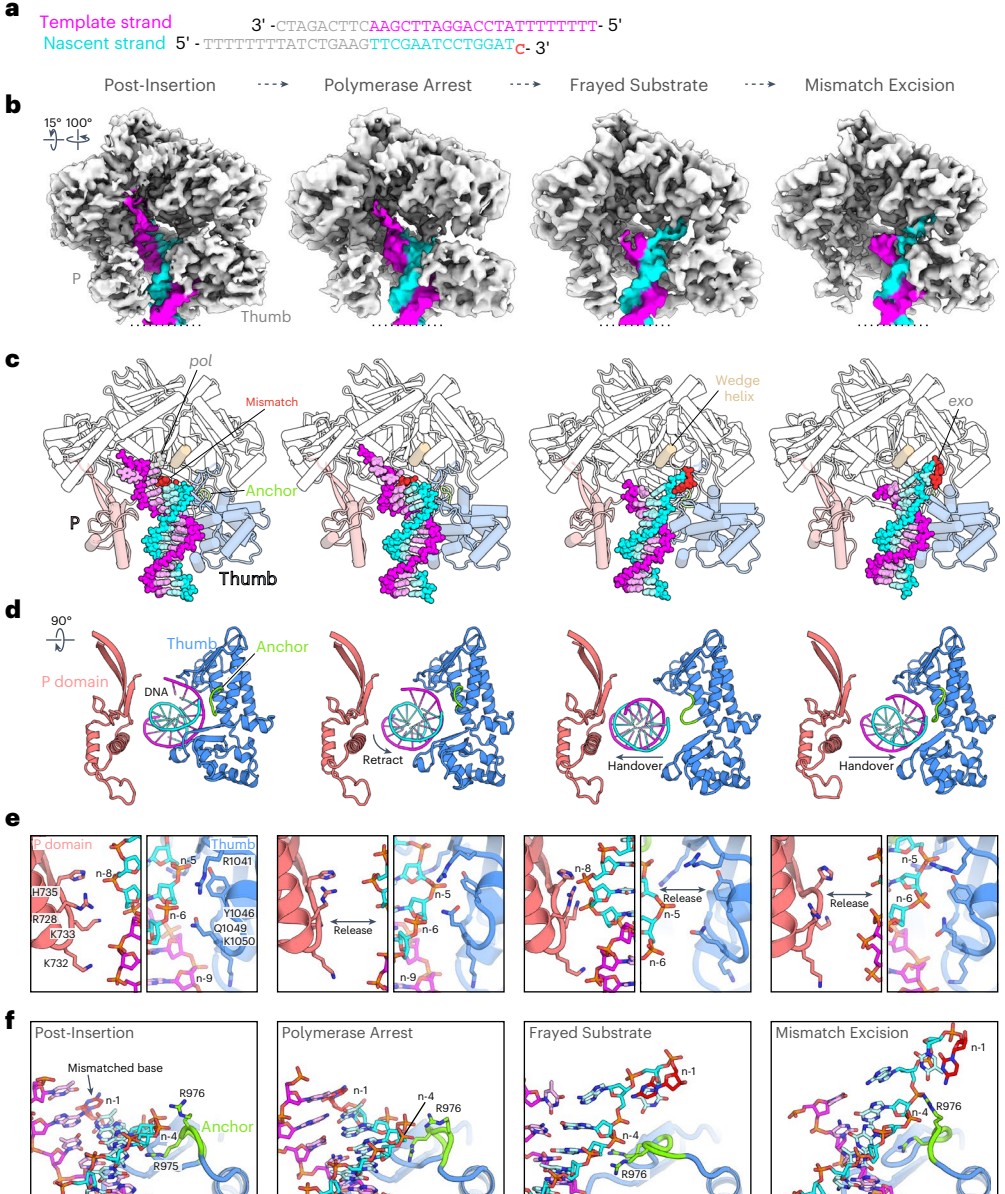

**Fig. 3 | Cryo-EM structures of Pol ε at a T–C-mismatched DNA substrate.**
**a**, DNA oligonucleotide pair that was used as the substrate for reconstitution.
**b–f**, The four distinct conformational states that were observed for Pol ε at
mismatched DNA. Shown are the cryo-EM density maps (**b**) and corresponding
atomic models underneath (**c–f**). The locations of active sites for polymerization
(*pol*) and proofreading (*exo*) are indicated in the complete ribbon models;
the mismatched nucleotide is colored in red (**c**). A top view (rotation symbol

relative to **c**, showing only indicated components) illustrates the Thumb domain
movement, rearrangements in the Anchor loop and the handover of the DNA
substrate between the Thumb and P domains (**d**). Close-up views show the
change in the interfaces between the DNA substrate and the P (salmon) and
Thumb (blue) domains (**e**) and between the DNA substrate and the Anchor loop
(green) (**f**). The DNA and coordinating side-chain residues of Pol ε are shown as
sticks; the mismatched nucleotide is colored red.

the reported Pol ε cat apo structure[35]. The DNA moves in conjunction
with the Thumb domain, maintaining its extensive coordination at
the Thumb's surface but disengaging its contacts with the P domain
(Fig. 3e). As a result, the 3′ end of the nascent strand is retracted
from the *pol* site into a position that is incompatible with further exten-
sion (Fig. 5b).

The conformational transition from the Post-Insertion to the
Arrest state could be promoted by several consequences of the incor-
porated mismatch. Firstly, the backbone distortion in the postinsertion
site compromises optimal base stacking with the incoming nucleotide,
likely decreasing its affinity[30]. In the absence of the incoming dNTP,
which bridges DNA interactions at the Palm and Finger domains, the
DNA remains almost exclusively coordinated at the Thumb domain

and would, therefore, be highly receptive to its movements. This is
supported by the observation that switching from polymerization to
editing activity becomes less likely with increasing nucleotide con-
centrations[48] and is also consistent with the absence of an incoming
nucleotide in the *pol* site in our Arrest state.

Secondly, the partial release of the interaction network in the
domain interface between the Palm and Thumb, as described above
(Fig. 4e), relaxes restraints at the base of the Thumb, which may facili-
tate the domain's rotating conformational change. Consistently, the
interactions at the base of the Thumb are further released in the Arrest
state, resembling the apo conformation of Pol ε in which the Thumb
takes a similar outward-rotated position[35]. Taken together, we hypoth-
esize that the described changes in the microenvironment surrounding

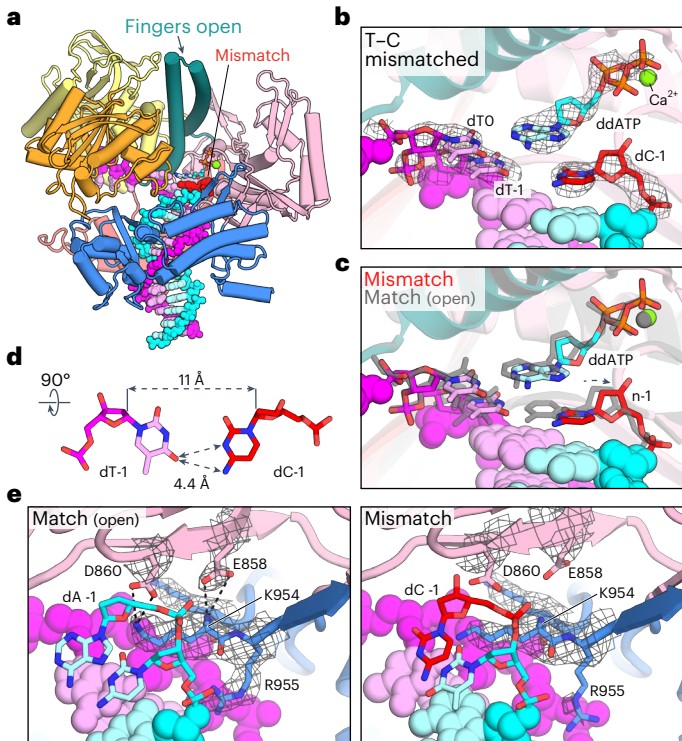

**Fig. 4 | Detection of a mismatched base pair located in the postinsertion site.** **a**, Ribbon model of Pol ε cat in complex with mismatched substrate DNA and incoming nucleotide in the replication conformer. **b**, *pol* active site of Pol ε cat with a T–C mismatch at the postinsertion site. The cryo-EM density is shown as a mesh around the incoming ddATP and its templating nucleotide dT0, as well as for the mismatched base pair at the postinsertion site marked as dT-1 and dC-1. **c**, Superposition of the *pol* active site with a T–C-mismatched (as in **b**) or T–A-matched (gray) primer terminus in the postinsertion site. **d**, The mismatched T–C base pair in the postinsertion site. Dashed arrows demark distances (in Å) between indicated atoms. Here, distances between the nucleobase faces and between respective C1′ atoms are shown. **e**, View of the 3′-terminal nucleotides of the nascent strand in the postinsertion (n-1) site, matched (left) or mismatched (right). Cryo-EM density is shown as a mesh around indicated side-chain residues, which are shown as sticks. Dashed lines indicate electrostatic and hydrogen-bond interactions between carboxyl groups of E858 and D860 in the Pol ε Palm domain and K954 of the Thumb domain.

the penultimate base pair lead Pol ε to remove the DNA from the *pol* site and enter a state of arrested polymerase activity in response to mismatch incorporation.

## DNA handover to the P domain and 3′-end fraying
For proofreading, the 3′ end of the nascent strand needs to be unpaired from the template strand for the terminal base to be accommodated in the *exo* site. Through a small torsion, the Thumb motion described above not only moves but also tilts the domain's DNA-binding surface, with a larger distance traversed at the sites that coordinate the nascent strand than the template strand. R1041, Y1046 and Q1049 move ~5 Å and pull the backbone of the nascent strand, coordinated at positions n-5 and n-6, by a corresponding distance. The template strand at positions n-7 to n-9 and coordinating residues K1050 and R1136 only traverse 2–3 Å. This torsion in the Thumb domain tilts the DNA double helix and increases the bend in the DNA path at positions n-3 to n-5. Resulting distortions in the DNA backbone and between optimally paired and stacked bases potentially weaken the duplex region upstream of the mismatched terminus to promote strand melting and proofreading (Fig. 5c).

In the Frayed Substrate state, we observe cryo-EM density at the junction of the template and nascent DNA strands that can

accommodate two bases of single-stranded DNA of the nascent strand at the entry cavity of the Exo domain (Fig. 6a,b). The atomic model of the Frayed state shows how the terminal bases of the nascent strand are loosely held at the Exo domain near residues N363, F366, H422 and N423 (Fig. 6b,c) and traject away from the template strand, from which they are physically separated by α-helix 3 of the Exo domain ('wedge helix'; Figs. 3c and 6a,b) that obscures the direct path between the *pol* and *exo* sites. The upstream portion of the dsDNA is released from its interactions at the tip of the Thumb domain (R1041 and Q1049 at nascent strand n-5 and n-6 and K1050 at template position n-9) and moves away from the Thumb and toward the P domain (Fig. 3c–e). In this configuration, the coordination of DNA at the P domain is more extensive than we observe in structures with matched DNA, with the backbone of the nascent strand held tightly against the positively charged surface of the P domain formed by residues R728, K733 and H735 (Fig. 3e). This 'handover' (Fig. 3d) of the DNA from the Thumb to the P domain could be initiated by the Thumb's tilting DNA-binding surface during the transition toward the Arrest state, which the DNA compensates for through an in-plane tilt that brings the upstream region closer to the P domain (Fig. 5c). Consistently, the Thumb further continues this tilt during the transition to the Frayed state. Positions n-3 and n-4 of the nascent strand remain coordinated by a flexible loop at positions 970–980 of the Thumb domain that harbors three positively charged residues K974, R975 and R976 (Fig. 3f). This 'Anchor loop' undergoes large rearrangements between the described states along the activity switching pathway to continuously coordinate the nascent strand at these positions. The uninterrupted anchoring may aid the melting of the terminal bases of the nascent strand from the template and guide the liberated 3′ end underneath the wedge helix during the DNA handover (Fig. 3f). Consistent with the Anchor loop having a critical role in activity switching, its double-arginine motif is conserved in Pol δ and the arginine in yeast Pol ε equivalent to human R975 was found to be critical for processive activity switching during proofreading[31] (Extended Data Fig. 4).

## Backtracking positions the *exo* site for mismatch excision
Despite being melted from the template strand and held at the surface of the Exo domain, the terminal erroneous base is still located more than 10 Å from the *exo* site in the Frayed state (Fig. 6c). In the fourth structure, the Mismatch Excision state, the DNA double helix has moved back to the Thumb domain where it is coordinated in a 1-bp-backtracked position compared to the Arrest state (Fig. 3c,e). The anew handover of the DNA double helix, now from the P domain back to the Thumb (Fig. 3d), coincides with further unwinding of the junction between nascent and template strands. We observe continuous cryo-EM density along the entry of the Exo domain and toward the buried active site that can accommodate the three terminal nucleotides of the nascent strand, with the erroneous base positioned at the *exo* site where the phosphate backbone of the terminal nucleotide is aligned with the catalytic residues of exonuclease motifs I, II and III (Fig. 6d–f). The Mismatch Excision conformer of Pol ε closely resembles previously reported structures of phage RB69 and *Pyrococcus abyssi* Pols[49,50].

## PCNA during proofreading
The conformational landscape suggested by our structures during the switch from polymerase to exonuclease activities involves rigid-body movements of the Thumb and P domains. Although the main interaction between Pol ε and PCNA is formed by the PIP box, the Thumb and P domains form direct contacts with the remaining protomers, which raises the question of whether these interfaces can accommodate the domain movements during proofreading. The Thumb forms the smallest of the contacts between the two proteins and the interface does not involve obvious interactions between key side-chain residues. Moreover, the large outward rotation of the Thumb during activity switching is directed parallel to the interface with PCNA, which does

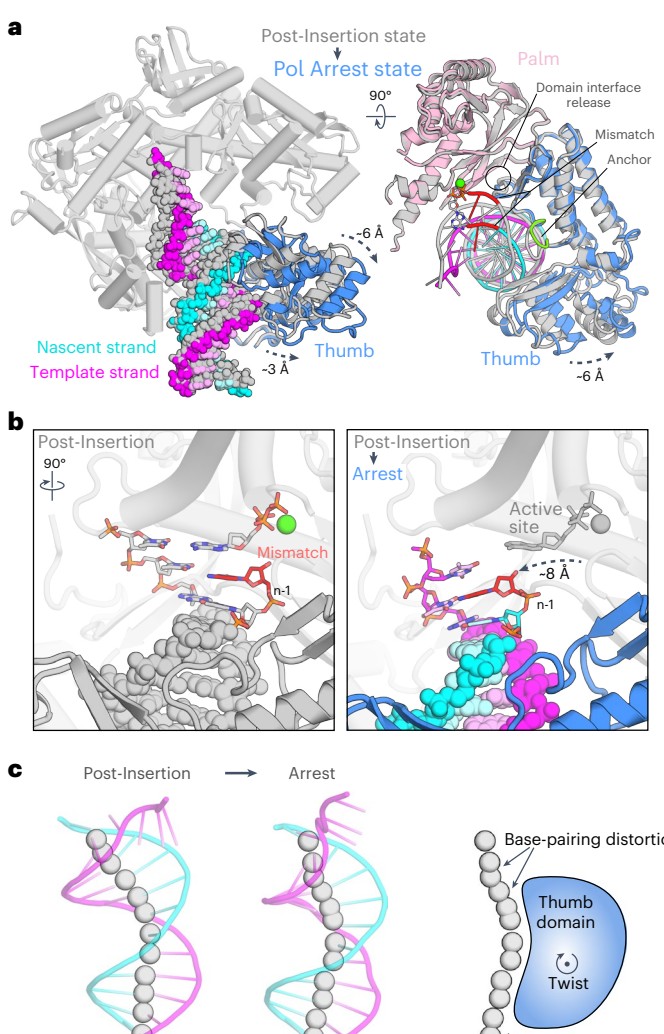

**Fig. 5 | Retrieval of erroneous DNA from the *pol* active site. a**, Superposition of Pol ε cat in the Post-Insertion (gray) and Pol Arrest (colorized) states. Dashed arrows indicate the outward opening movement of the Thumb domain. **b**, View of the *pol* active site comparing the Post-Insertion (left) and the Arrest (right) states. The incoming nucleotide is absent in the structure of the Arrest state but is depicted in both panels for reference. The rotation symbol indicates the view relative to **a**, left. **c**, Path of the DNA double helix in the two states shown as a cartoon representation of the atomic model and as spheres at the centroids between the backbone phosphates of the paired strands. The Thumb domain is shown as a simplified cartoon and its twisting movement is indicated by the rotation symbol. The DNA distortion in positions n-2 to n-5 (with n-1 being the terminal mismatch) in the Arrest state is indicated and becomes visible from overlapping spheres and the kinked trace of the centroids. The upstream portion of the DNA tilts away from the Thumb and toward the P domain.

not pose restraints to the Thumb's mobility. On the other hand, the P domain is more rigidly fixed at its contact with PCNA (Extended Data Fig. 2d) and we observe PCNA shifting together with the P domain to accommodate the outward movement of ~3.5 Å that we observe in the Frayed Substrate state.

The position of PCNA with respect to Pol ε in the replicating holoenzyme assembly appears very homogeneous, as revealed by masked 3D classification (Extended Data Fig. 5a), likely because of the tripod-like arrangement of the three contact sites. This is also true for the Frayed Substrate proofreading intermediate where DNA is coordinated at the P domain and the nearby PCNA protomer. In these states, we observe virtually complete PCNA occupancy and only

minor heterogeneity in its position (Extended Data Fig. 5). In contrast, PCNA shows a substantially higher degree of mobility in the states in which the P domain binds the DNA less strongly (Post-Insertion state) or not at all (Arrest and Mismatch Excision states). This implies that the stable position of PCNA with respect to Pol ε cat is supported and maintained by adjacent DNA coordination sites, whereas the position of PCNA is destabilized upon DNA release from the P domain. The 3D classification also revealed particle subsets from the Arrest and Mismatch Excision states that show PCNA tilted away from the Thumb and P domains, hinging around the interaction with the PIP box (Extended Data Fig. 5b). Similar rotational freedom around the axis of the PIP box interaction was observed for Pol δ–PCNA and could free up sites for PIP box interactions (for example, for the recruitment of factors for extrinsic proofreading)[13,51].

## Discussion

Our cryo-EM structures of the human leading-strand polymerase Pol ε in complex with PCNA and DNA reveal how Pol ε binds PCNA with its PIP box and forms additional contacts with unique features in the Thumb and P domains. Furthermore, we present a series of structures of Pol ε engaged at a mismatch-containing DNA that provide insights into proofreading and activity switching mechanisms.

In comparison to the lagging-strand polymerase Pol δ, which associates with PCNA through only one protomer to allow the additional recruitment of Okazaki fragment maturation factors such as FEN1 to an exposed PCNA protomer[13], Pol ε contacts all three PCNA protomers. Although the two additional contacts may only contribute marginally to the formation of the complex (disruption of the interaction at the PIP box is sufficient to abolish PCNA-dependent rate enhancement during DNA replication[52]), this tripartite interaction masks the additional recruitment sites on PCNA, which likely explains why the Pol ε–PCNA holoenzyme is impervious to other PIP-box-containing factors such as FEN1 or free Pol δ (refs. 22,52). Nonetheless, the masked PIP box interaction sites may become accessible upon polymerase arrest and during proofreading.

Although there is considerable flexibility between the Pol ε noncatalytic and catalytic domains, the catalytic domain has been observed in two distinct positions in CMG–Pol ε structures[35,36,53]. The PCNA interaction of the ternary complex that we report here is sterically compatible with both conformations and also with the recruitment of the Ctf18-1-8 module of the Ctf18–replication factor C clamp loader[35,54] (Extended Data Fig. 6).

A common characteristic among DNA Pols is the conformational change of the Fingers that is induced by the binding of a complementary dNTP in the *pol* site[28,41]. Because Finger closing is required for the extension reaction, the dynamics of the Finger-closing motion and distinct states along it provide putative checkpoints to verify correct base pairing and overall geometry of the incoming nucleotide to establish prechemistry fidelity[27,29]; the dynamics may additionally be sensitive toward the geometry of the DNA substrate and, thus, able to detect incorporated mismatches and initiate the switch to exonuclease activity for proofreading[55,56]. Consistently, we observe the Fingers of Pol ε in open, closed and ajar conformations when Pol ε is engaged at an error-free DNA substrate. With a T–C mismatch at the terminus of the nascent strand, we observe Pol ε only in the open conformation despite the presence of a complementary nucleotide triphosphate at the identical concentration, suggesting a role of the Finger-closing conformational change in postchemistry fidelity. Other differences in sample preparation, such as the use of $Ca^{2+}$ as the bivalent chelating metal and the presence of a 3′-OH at the primer terminus, are unlikely to interfere with Finger closing[18,44,57,58]. The otherwise high structural similarity of Pol ε cat when bound to either matched or mismatched DNA substrates is consistent with a crystallographic study of the Pol α ternary complex at an inserted T–C mismatch[30]. Strikingly, Pol α adopts the closed conformation in the crystal structure, suggesting

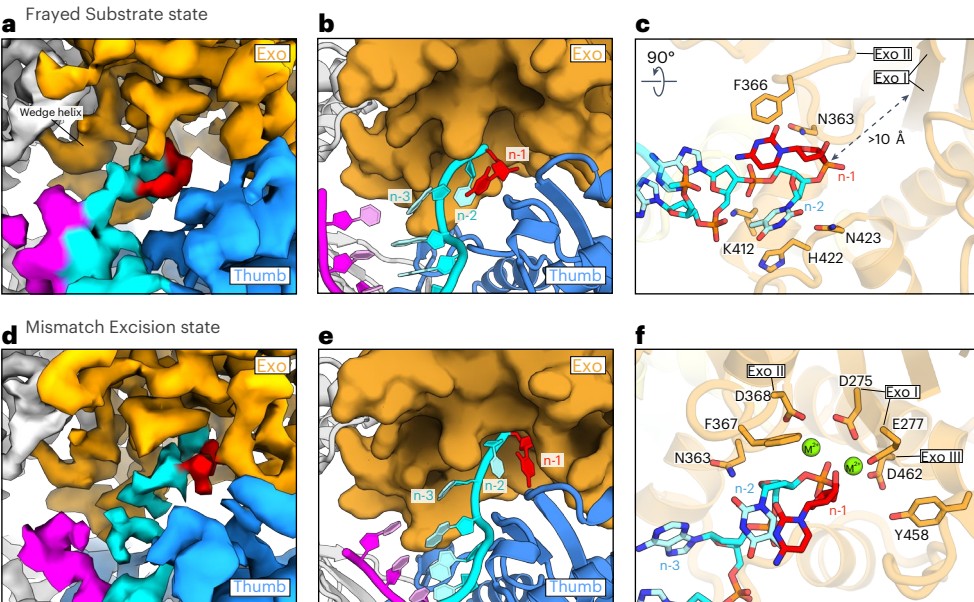

**Fig. 6 | Substrate fraying and subsequent backtracking bring the erroneous nucleotide toward the exonuclease active site for excision. a**, Surface view of the cryo-EM density map for Pol ε in the Frayed Substrate state shown at the entry site of the Exo domain. Thumb domain, blue; Exo domain, orange; template DNA strand, magenta; nascent DNA strand, cyan. The misincorporated nucleotide at the terminus of the nascent strand is colored in red. **b**, Ribbon view of the atomic model for Pol ε cat in the Frayed Substrate state shown for the same area as in **a**. The Exo domain is shown in surface representation. **c**, View of the 3′-terminal nucleotides of the nascent strand in the Frayed Substrate state. Side-chain residues contacting the nascent strand at the entry cavity of the Exo domain are shown as sticks. A dashed arrow indicates the distance between the 3′-terminal nucleotide and the *exo* active site, marked at Exo I and Exo II motifs. The rotation symbol in this figure indicates the view relative to **b. d**, Surface view of the cryo-EM density map for Pol ε in the Mismatch Excision state. Colors and view are as in **a**. **e**, Ribbon view of the atomic model for Pol ε in the Mismatch Excision state shown for the same area as in **d**. For visibility, the side chain of F285 was removed from the surface representation of the Exo domain. **f**, View of the 3′-terminal nucleotides of the nascent strand in the Mismatch Excision state. DNA-contacting and catalytic amino acid side-chain residues are shown as sticks. Residues of exonuclease motifs I, II and III are marked. Spheres represent bivalent catalytic metal ions.

additional quality control mechanisms that contribute to the detection of an incorporated error and the prevention of further extension. Base stacking between the incoming nucleotide and the penultimate nucleobase in the postinsertion site was observed to contribute substantially to the incorporation efficiency[59]. Our structure illustrates how an inserted T−C mismatch compromises base stacking with the incoming nucleotide, which offers a possible explanation for decreased dNTP affinity after a misinsertion[30] and, thus, disfavored Finger closing. Additionally, the backbone distortion is consistent with a proposed mechanism that the altered angle of nucleophilic attack by the 3′ end slows extension[30]. All three consequences of the misinserted base (disfavored Finger closing, decreased dNTP affinity and altered angle of nucleophilic attack) are likely to occur additively to serve postinsertion error detection, prevent further extension and trigger proofreading by switching to 3′-exonuclease activity[27,58].

Our structures show that a misincorporated nucleotide located in the postinsertion site can also destabilize the interface between the Thumb and Palm domains. This mobilizes the Thumb domain and sets the stage for activity switching and proofreading in Pol ε, for which we propose a three-step mechanism that is based on two intermediate conformers. The mechanism involves repeated handover of the DNA double helix inside the hand architecture of Pol ε that is coupled to forward translocation and backtracking movements, while the continuous coordination of the nascent DNA strand near its terminus guides the DNA substrate between *pol* and *exo* active sites. The mismatched substrate triggers a large movement of the Thumb domain that expulses the DNA substrate from the *pol* site and shifts Pol ε into a distinct Polymerase Arrest state in which further extension past the mismatch is prevented and the junction between template and nascent strands becomes destabilized through backbone distortion. In the subsequent transition into the Frayed Substrate state,

the DNA double helix is released from the tip of the Thumb and handed over to the P domain, while transient forward translocation allows the erroneous 3′ end to pass underneath the separating wedge helix and become physically separated from the template strand. The third step marks the transition to the Mismatch Excision conformer, in which the DNA substrate is channeled deeper into the Exo domain cavity. The transition involves a handover of the DNA from the P domain back to the Thumb combined with a 1-bp backtracking movement of Pol ε, which results in the separation of a third base pair between nascent strand and template. Unlike most B-family Pols, Pol ε lacks the extended β-hairpin loop that was proposed to stabilize the nascent strand in the *exo* site and aid in processivity during proofreading (Extended Data Fig. 2h)[17,49,60]. Instead, our data reveal a substantial involvement of the unique P domain that, together with the Thumb, channels the erroneous base toward the *exo* site to position Pol ε for mismatch excision.

The mechanism we propose for activity switching and proofreading is corroborated by many previous functional and biochemical studies[16,17,25,26,31,47,48,52,61–64] (Extended Data Fig. 4) and is consistent with a prevailing 'frayed primer template' model for A-family Pols[32,33]. Our work marks the first structural characterization of proofreading in a eukaryotic B-family Pol and deepens our mechanistic understanding of the unique Pol that catalyzes high-fidelity leading-strand DNA replication.

## Online content

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

## Methods

### Molecular cloning

The D275A;E277A point substitutions in the POLE1 exonuclease active site were introduced by excision and replacement of the 1,007-bp segment between BamHI and ApaI restriction sites in the POLE1-pAceBac1 plasmid[11]. The replacement insert was produced in a two-step assembly PCR with flanking primers AAATGATAACCATCTCGC and GTACAGGTAGTAGGTGGCCAC and primers for directed mutagenesis GGCTTTCGCCATCGCGACTACCAAGCTGCCCCTGAAG and GGTAGTCGCGATG GCGAAAGCCAACACCACAGGATC. The calmodulin-binding protein tag on POLE3 was removed from POLE3-pAceBac1 (ref. 11) by plasmid PCR with primers flanking the sequence encoding the tag (forward, GAAGAGGTC GACAACTAAGGTACCTCTAGAGCCTGCAGTCTCG; reverse, CTAGAGGTACCTT AGTTGTCGACCTCTTCTTCCTCGTTTTGTTCTTCC) to generate vector DNA that was then transformed into NEB 5-alpha competent *Escherichia coli*. Generated plasmids were verified by Sanger sequencing.

### Protein purification

PCNA was expressed in *E. coli* BL21(DE3) Rosetta and purified as described previously[36]. Wild-type and exonuclease-deficient Pol ε were purified following an adaptation of a previously published protocol[11,63]. Cells from a 0.5 l culture, grown as previously described[36], were resuspended in 100 ml of lysis buffer (100 mM Tris-acetate (pH 7.5), 250 mM sodium acetate, 5% glycerol, 1 mM EDTA and 0.5 mM TCEP) + protease inhibitors (cOmplete, EDTA-free, Roche; one tablet per 50 ml of buffer). Cells were lysed by Dounce homogenization and insoluble material was cleared by centrifugation (235,000$g$ at 4 °C for 45 min). In two batches of 50 ml, the cleared lysate was applied to a 5 ml HiTrap heparin column (Cytiva) equilibrated in 25 mM HEPES–NaOH (pH 7.6), 10% glycerol, 1 mM EDTA, 0.005% NP-40-S and 280 mM sodium acetate (buffer B + 280 mM sodium acetate) and eluted across a 20-column-volume (CV) gradient to buffer B + 1.2 M sodium acetate. Pol ε-containing fractions (as determined by SDS–PAGE) were pooled, diluted threefold in buffer B + 50 mM sodium acetate and applied to a 5 ml HiTrap Q FF column (Cytiva) equilibrated in buffer B + 280 mM sodium acetate. Bound protein was eluted with a 20-CV gradient to buffer B + 910 mM sodium acetate. Peak fractions were pooled, diluted 2.5-fold in buffer B + 50 mM sodium acetate, applied to a 1 ml MonoQ column (Cytiva) equilibrated in buffer B + 280 mM sodium acetate and eluted with a 30-CV gradient to buffer B + 1.2 M sodium acetate. Peak fractions were pooled, concentrated to approximately 600 μl and applied to a Superdex 200 increase 10/300 column (Cytiva) equilibrated in 25 mM HEPES–NaOH (pH 7.6), 10% glycerol, 1 mM EDTA, 0.005% NP-40-S, 0.5 mM TCEP and 500 mM sodium acetate. In cases where peak fractions from gel filtration still contained contaminating protein detectable on Coomassie-stained SDS–PAGE, the sample was diluted fivefold in buffer B + 50 mM, loaded on a 1 ml MonoS column (Cytiva) equilibrated in B + 100 mM sodium acetate and eluted with a 15-CV gradient to buffer B + 970 mM sodium acetate. Peak fractions were pooled, frozen in liquid nitrogen and stored at −80 °C. Protein yield was approximately 0.8 mg from 0.5 l of cell culture.

### DNA constructs

DNA oligomers (Integrated DNA Technologies) were dissolved in Tris-EDTA at 200 μM. Complementary oligos were mixed at an equimolar ratio in 25 mM HEPES–NaOH (pH 7.5), 150 mM sodium acetate, 2 mM magnesium acetate (pair 1) or calcium acetate (pair 2) and annealed by gradual cooling from 80 °C to room temperature.

Oligomer sequences were as follows:

1. Pair 1 (match)
   (a) Template, 5′-AAGGCTGAACGAATTGGTGAGGGTTGGGAAG TGGAAGG-3′
   (b) Primer, 5′-biotin-CCTTCCACTTCCCAACCCTCACC-3′

2. Pair 2 (mismatch)

   (a) Oligo 1, 5′-T*T*T*TTTTTATCCAGGATTCGAACTTCAGAT*C-3′
   (b) Oligo 2, 5′-T*T*T*TTTTTATCTGAAGTTCGAATCCTGGAT*C-3′, where * denotes a nonbridging phosphorothioate modification in the phosphate backbone

### Preparation of complexes for cryo-EM

**Pol ε D275A;E277A with PCNA on matched DNA in the presence of ddATP.** Exonuclease-deficient Pol ε was mixed with 1.5-fold molar excess DNA (pair 1) and diluted with a buffer containing 25 mM HEPES–NaOH pH 7.5, 50 mM sodium acetate, 0.5 mM TCEP, 10 mM magnesium acetate (binding buffer + 50 mM sodium acetate) to reach an ionic strength approximately corresponding to binding buffer + 100 mM sodium acetate. Mixtures were incubated on ice for 1 h. PCNA was separately adjusted to a matching ionic strength as above before mixing with Pol ε and DNA in 1.5-fold molar excess over Pol ε. After 45 min of further incubation on ice, the mixtures were concentrated to approximately 60 μl using an Amicon ultra centrifugal filter with a 30 kDa molecular weight cutoff. Monovalent streptavidin was added at twofold molar excess and ddATP was added to 250 μM concentration. The mixture was centrifuged at 21,000$g$ for 5 min before injection onto a 2.4 ml S200 increase 3.2/300 column equilibrated with binding buffer + 100 mM sodium acetate. Peak fractions were identified by SDS–PAGE, concentrated to ~0.55 mg ml$^{-1}$ and supplemented with an additional 125 μM ddATP. A 12 μl aliquot was supplemented with 3 μl of 0.05% glutaraldehyde and incubated on ice for 10 min. A 3.5 μl sample was applied to freshly glow-discharged (PELCO easiGlow 91000, glow for 45 s at 25 mA and 0.39 mbar) UltrAuFoil R1.2/1.3, manually blotted on the top face for 2 s and immediately plunged into liquid ethane at liquid nitrogen temperature. This sample was separately analyzed in negative-stain room temperature transmission electron microscopy, as described below. Separately, for samples with CHAPSO detergent, Pol ε was mixed with 1.3-fold molar excess DNA, an equimolar amount of PCNA and ddATP as described above. After incubation on ice, the sample was concentrated to 4.3 mg ml$^{-1}$ (approximately 10 μM). CHAPSO was added to a final concentration of 8 mM immediately before vitrification on UltrAuFoil R1.2/1.3 grids as described above. No streptavidin was added to the sample with CHAPSO detergent.

**Wild-type Pol ε with PCNA on mismatched DNA in the presence of ddATP.** For the reconstitution of Pol ε−PCNA mismatched DNA, we used wild-type Pol ε instead of the exonuclease-inactive variant and substituted Ca$^{2+}$ for Mg$^{2+}$ in the reconstitution buffers. Pol ε was mixed with 1.5-fold molar excess mismatch DNA (pair 2) and diluted with binding buffer + 50 mM sodium acetate as described above, substituting calcium acetate for magnesium acetate. After incubation on ice for 15 min, PCNA was added and incubated for a further 20 min on ice. The mixtures were supplemented with ddATP and subjected to size-exclusion chromatography as described above; peak fractions were pooled, concentrated to approximately 40 μl (~0.44 mg ml$^{-1}$) and supplemented with 125 μM ddATP. A 12 μl aliquot was mixed with 3 μl of 0.05% glutaraldehyde and incubated on ice for 6 min. A 3.5 μl sample was applied to freshly glow-discharged (Edwards S150B for 50 s at 33 mA) UltrAuFoil R1.2/1.3, manually blotted on the top face for 2 s and immediately plunged into liquid ethane at liquid nitrogen temperature. Separately, for samples with CHAPSO detergent, Pol ε was mixed with twofold molar excess DNA and an equimolar amount of PCNA in calcium-substituted binding buffer as above. The mixture was incubated at 30 °C for 15 min, returned to ice for an additional 30 min and concentrated to 3.5 mg ml$^{-1}$ (approximately 8 μM). For plunge-freezing, the sample was supplemented with 0.015% glutaraldehyde and incubated on ice for 5 min. CHAPSO was added to a final concentration of 8 mM immediately before vitrification on UltrAuFoil R1.2/1.3 grids as described above.

## Cryo-EM data collection

**Pol ε D275A;E277A with PCNA on matched DNA in the presence of ddATP (dataset 1).** A dataset of 13,522 multi-frame movies was acquired using a 300 kV TFS Titan Krios microscope, equipped with a Gatan K3 direct electron detector (electron counting mode; 20 eV slit width for the BioQuantum energy filter). Data collection was controlled by EPU automated acquisition software (Thermo Fisher) in 'faster acquisition' mode (AFIS). Images were recorded in super-resolution mode bin 2 at an effective pixel size of 0.73 Å per pixel over a defocus range of −0.8 to −3.0 μm. Movies were dose-fractionated into 39 fractions over a 1.2 s exposure time with a total dose of 40.08 e− per Å$^2$.

**Pol ε D275A;E277A with PCNA on matched DNA in the presence of ddATP and 8 mM CHAPSO (dataset 2).** A dataset of 12,784 multi-frame movies was acquired using a 300 kV TFS Titan Krios microscope equipped with a Falcon 4i direct electron detector operated in electron counting mode using EPU with AFIS enabled. Images were recorded over a defocus range of −0.8 to −3.0 μm, at an effective pixel size of 0.824 Å per pixel and over a 2.67 s exposure time with a total dose of 40.18 e− per Å$^2$.

**Wild-type Pol ε with PCNA on mismatched DNA in the presence of ddATP (dataset 3).** A dataset of 9,659 multi-frame movies was acquired using a 300 kV TFS Titan Krios microscope equipped with a Gatan K3 direct electron detector (operated as above). Images were recorded in super-resolution mode bin 2 at an effective pixel size of 0.73 Å per pixel over a defocus range of −1.2 to −3.0 μm. Movies were dose-fractionated into 39 fractions over a 1.3 s exposure time with a total dose of 36.4 e− per Å$^2$.

**Wild-type Pol ε with PCNA on mismatched DNA in the presence of 8 mM CHAPSO (dataset 4).** A dataset of 13,159 multi-frame movies was acquired using a 300 kV TFS Titan Krios microscope equipped with a Gatan K3 direct electron detector detector (operated as above). Images were recorded in super-resolution mode bin 2 at an effective pixel size of 0.73 Å per pixel over a defocus range of −1.2 to −3.0 μm. Movies were dose-fractionated into 39 fractions over a 1.4 s exposure time with a total dose of 39.9 e− per Å$^2$.

## Cryo-EM image processing

**Pol ε cat–PCNA on matched DNA.** The data processing pipeline outlined here is shown as a schematic in Extended Data Fig. 1f. The 13,522 and 12,784 collected movies of datasets 1 and 2 were imported in CryoSPARC (version 4)[65] and subjected to patch motion correction and patch contrast transfer function (CTF) estimation. Micrographs of poor quality were excluded from further processing. From the remaining 5,723 and 10,857 micrographs, particles were picked using the Blob Picker and ~2.6 million and ~3.9 million particles were extracted, respectively. The two particle sets were separately subjected to 2D classification, after which subsets of 300,000 particles from classes with good 2D class averages were used for ab initio reconstruction. The generated maps were then provided as references for separate heterogeneous refinements of all extracted particles, from which the best class (displaying clear features for Pol ε cat, PCNA and DNA) was selected, respectively. During two additional rounds of classification, the best-quality particles of Pol ε–PCNA bound to the 5′ overhang of the DNA substrate were isolated, while lower-quality images and particles that had bound the biotinylated, blunt-ended side of the duplex were excluded. In total, 237,296 and 106,441 particles from the two datasets, respectively, were re-extracted and Fourier-cropped to achieve the same particle box and pixel sizes, combined and subjected to homogeneous refinement. Merging of the two datasets substantially improved the resolution of the map and reduced anisotropy from preferred particle orientation in the absence of CHAPSO detergent (compare refined maps before and after merging in Extended Data Fig. 1f). Next, the merged particles were aligned in a local refinement providing a focus mask around Pol ε cat and then classified in 3D into ten classes, providing the same focus mask and without realigning particles. Particle classes were combined on the basis of the state of the Finger domain and individually refined using a focus mask around Pol ε cat (local refinement) or subjected to non-uniform refinement[66]. Statistics for refinement and reconstructions are summarized in Table 1. Additionally, Extended Data Fig. 1g–j shows the highest populated 2D class averages generated from the final particle set of the open conformation, the final reconstruction in the open conformation colored by local resolution, Fourier shell correlation (FSC) plots for the three final maps from non-uniform refinement and example cryo-EM density (mesh) for the second α-helix in the Finger domain for the three states.

**Pol ε cat–PCNA on mismatched DNA.** The data processing pipeline outlined here is shown as a schematic in Extended Data Fig. 3a. The collected movies from datasets 3 and 4 were motion-corrected and dose-weighted using RELION's implementation of a MotionCor2-like program[67,68]. Micrographs were then imported in CryoSPARC (version 4), subjected to patch CTF estimation, particle picking and extraction and heterogeneous refinements as described above. Example micrographs are displayed in Extended Data Fig. 3b,c. The best particle subsets containing Pol ε cat, PCNA and DNA (571,939 and 889,791 particles, respectively) were combined and, as above, first refined by homogeneous refinement, followed by a local refinement providing a focus mask around the Pol ε cat (Extended Data Fig. 3d, mask I).

Next, the refined particles were classified in 3D into 12 classes providing a focus mask around the DNA and Exo and Thumb domains of Pol ε (Extended Data Fig. 3d, mask II). The four largest of 12 total classes (based on the number of assigned particles) contained the particles of the highest quality, while the subsets of the remaining classes could not be separately refined to high resolution. The same 3D classification strategy, but preceded by a heterogeneous refinement (three classes) and selecting the highest resolution particles for subclassification, resulted in a comparable result with the four best classes (of eight total classes here) representing the Pol ε–DNA complex in four distinct states.

Duplicate particles were removed on the basis of refined particle locations, and the four particle subsets were separately processed further as follows: All further 3D classification steps were performed without realigning particles and with focus masks around the entire Pol ε cat (Extended Data Fig. 3d, mask I), around the Thumb and Exo domains and the bound DNA (Extended Data Fig. 3d, mask II), around the Thumb domain and the DNA (Extended Data Fig. 3d, mask III), only around the DNA (Extended Data Fig. 3d, mask IV) or around the area encompassing the junction of the DNA substrate and the *exo* site (Extended Data Fig. 3d, mask V). After classification, the individual particle subsets were refined separately (using the local refinement job in CryoSPARC and providing mask I) and classes were selected on the basis of the quality of the resulting map ('best class') or the number of assigned particles ('largest class').

Particles in the replication-like conformer (the Post-Insertion state) were subjected to one final round of 3D classification enabling the 'force hard classification' option and with the same mask around Pol ε cat. Particles in classes of poor quality were excluded from the final refinement.

Particles in the Arrest state were subjected to signal subtraction removing the signal for PCNA. Three iterative rounds of 3D classification with the 'force hard classification' option enabled were performed to remove poor-quality particles that were sorted into classes of smaller population. To improve the signal for the DNA and the Thumb domain, one final round of 3D classification with a mask around these features (mask III) was performed. The particles of the best class were selected and refined after reverting to the original, unsubtracted particles to obtain the final map.

The particles in the Frayed Substrate state were subjected to two parallel 3D classification jobs with a focus mask around the Thumb and Exo domains and the bound DNA (mask II) and particles of poor quality were discarded. The remaining particles were subjected to one additional round of 3D classification providing the same focus mask, after which particle classes were selected that displayed the best density for the nascent strand at the DNA junction and for the Anchor loop, as well as side-chain density for nearby residues such as F366. The selected subset was refined as described above.

Particles in the Mismatch Excision state were first processed using two different strategies. On one hand, particles with poor signal for the DNA double helix were excluded after one round of focused 3D classification with mask V. Then, the signal for PCNA was subtracted and particles were 3D classified with focus mask II (around the Thumb domain, Exo domain and DNA), after which the particles of the two best classes were combined and refined. On the other hand, the signal for PCNA was subtracted from all particles in the initial set of the Mismatch Excision state and the particles were subjected to 3D classification with a focus mask around the DNA (mask IV). The 3D class that showed good density for the DNA double helix in the same register as the refined map in strategy 1 was selected and combined with the selected particles from strategy 1. The combined particles were further classified in two iterative rounds, first providing mask IV and selecting the class with the best signal for the DNA double helix and then once with mask V, after which the largest particle class was selected, reverted to the original unsubtracted particles and refined to obtain the final map.

**Analysis of component flexibility between PCNA and Pol ε.** Particles sets of the Pol ε–DNA–PCNA complexes in the different states were separately subjected to global homogenous refinements followed by 3D classification (without realigning particles) into six classes and with a focus mask around PCNA. The 3D class averages were filtered to 5 Å resolution.

### Molecular modeling

**Pol ε cat–PCNA on matched DNA.** An initial atomic model was obtained from an AlphaFold-Multimer prediction providing POLE1 residues 1–1250 and three protomer copies of human PCNA[34]. This model was rigid-body fit into the final map for the Pol ε–PCNA complex at matched DNA in the open conformation using UCSF Chimera[69]. The DNA substrate and ddATP ligand were modeled in Coot[70]. The fit-to-density and model geometry were subsequently refined using ISOLDE inside ChimeraX[71] and with previously reported models for human PCNA and *S. cerevisiae* Pol ε as visual references[17,20]. PCNA closely resembles the previously reported crystal structure with an r.m.s.d. of ~1.3 Å for 750 pairs of Cα atoms (Protein Data Bank (PDB) 1AXC)[20]. At the CysX motif (cysteine residues 651, 654, 663 and 747), the electron density and the conformation of the surrounding area support the coordination of an iron–sulfur [4Fe–4S] cluster consistent with previous reports[38,39].

For structures in ajar and closed conformations, the generated model was rigid-body fit into the respective final maps and adjusted as above. All models were refined in an all-atom simulation in ISOLDE, applying distance restraints for all Watson–Crick base pairs and adaptive distance restraints ($\kappa = 5$) for the DNA substrate in the region upstream of n-7. The resulting models were refined using phenix. real_space_refine[72] against the final maps from non-uniform refinement (Extended Data Fig. 1f), providing ligand restraints generated in AceDRG[73]. Model validation was carried out using MolProbity[74] inside Phenix after a single round of atomic displacement parameter refinement against the respective final maps and is summarized in Table 1.

**Pol ε cat on mismatched DNA.** The model for Pol ε in the open conformation was rigid-body fit into the map for Pol ε bound to the mismatched DNA substrate in the replication conformer. The DNA

sequence was adjusted in Coot and the fit-to-density of the model was refined using ISOLDE. Because the best maps were obtained through 'focused' refinements with a mask around Pol ε cat that excludes PCNA and the DNA duplex portion upstream of n-14 (Extended Data Fig. 3c), those excluded regions were removed from the model. The positions of the DNA substrate in the three remaining proofreading conformers were determined as follows: For the Pol Arrest state, the complete model of the replication conformer was rigid-body fit into the final map and refined in an all-atom simulation using ISOLDE. For the Frayed Substrate and Mismatch Excision states, the terminal bases of the nascent strand were manually placed into the assigned density, while a separate model of ideal B-form DNA was rigid-body docked into the density for the upstream duplex portion before joining the DNA chains. The fit-to-density for all models was refined in all-atom simulations in ISOLDE, applying distance restraints for all Watson–Crick base pairs and adaptive distance restraints for the DNA substrate ($\kappa = 5$ in the region upstream of n-9 in the Post-Insertion state; $\kappa = 3$ upstream of n-9 and $\kappa = 0.5$ in the region n-9 to n-6 in the Arrest state; $\kappa = 5$ in the region upstream of n-4 in the Frayed Substrate state; $\kappa = 25$ in the region upstream of n-7 and $\kappa = 0.5$ in the region n-7 to n-4 in the Mismatch Excision state). The resulting models were refined using phenix. real_space_refine and validated as described above (Table 1).

The r.m.s.d. between atomic models was calculated using PyMOL version 2.3.4 (Schrödinger, www.pymol.org). Contact surface areas were determined using the 'protein interfaces, surfaces and assemblies' service at the European Bioinformatics Institute (www.ebi.ac.uk/pdbe/prot_int/pistart.html)[75].

### Negative-stain transmission electron microscopy and image processing

An aliquot of pooled and concentrated peak fractions from the Superdex 200 size-exclusion-purified complex (described above, ~0.55 mg ml⁻¹) was taken before the addition of a cross-linking agent. The sample was diluted 100-fold in binding buffer + 100 mM sodium acetate and 5 µl of the sample was applied to 3 nm carbon film on 400-mesh copper (Agar Scientific) freshly glow-discharged at 0.39 mbar and 30 mA for 40–50 s using a PELCO easiGlow. After incubation for 2 min, the grid surface was washed and stained by picking up three droplets (each 10 µl) of water and two drops of 2% uranyl acetate, blotting the grid on the side between drops. The second drop of uranyl acetate was incubated for 1 min and blotted and grids were dried overnight. Negative-stain samples were imaged on a 120-kV FEI Tecnai Spirit equipped with a Gatan Ultrascan 1000XP detector. A total of 50 micrographs were taken at ×26,000 magnification with a pixel size of 3.95 Å per pixel using Digital Micrograph camera software. Data from negative-stain EM were processed in RELION 4.0 (ref. 67). Particles were first picked with the Laplacian-of-Gaussian autopicker implementation inside RELION, extracted and classified in 2D. Favorable 2D class averages were selected and used for template-based autopicking in RELION on all micrographs. Extracted particles were subjected to two successive rounds of 2D classification.

### Reporting summary

Further information on research design is available in the Nature Portfolio Reporting Summary linked to this article.

## Data availability

Structure coordinates were deposited to the Research Collaboratory for Structural Bioinformatics PDB (https://www.rcsb.org/) and cryo-EM density maps used in model building were deposited to the EM Data Bank (https://www.ebi.ac.uk/pdbe/emdb) under the following accession codes: 9F6D and EMD-50222 (Pol ε–PCNA on matched DNA, open Finger conformation), 9F6E and EMD-50223 (Pol ε–PCNA on matched DNA, ajar Finger conformation), 9F6F and EMD-50224 (Pol ε–PCNA on matched DNA, closed Finger conformation), 9F6I and EMD-50225

(Pol ε–PCNA on mismatched DNA, Post-Insertion state), 9F6J and EMD-50226 (Pol ε–PCNA on mismatched DNA, Pol Arrest state), 9F6K and EMD-50227 (Pol ε–PCNA on mismatched DNA, Frayed Substrate state), and 9F6L and EMD-50228 (Pol ε–PCNA on mismatched DNA, Mismatch Excision state). Other data supporting the findings of this study are available from the corresponding author on request.

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

## Acknowledgements

We thank J. Shi for operation of the Medical Research Council (MRC) Laboratory of Molecular Biology (LMB) baculovirus facility and F. Fassetta (MRC LMB) for baculovirus generation and protein expression; M. Jenkyn-Bedford (MRC LMB) for recombinant monovalent streptavidin; S. Chen, G. Sharov, G. Cannone, A. Yeates and B. Ahsan for smooth running of the MRC LMB EM facility; and J. Grimmett, T. Darling and I. Clayson for maintenance of scientific computing facilities. We are grateful to K. Nguyen, S. Zhang, D. Czernecki and all members of the Yeeles lab for discussions and comments on the manuscript. This work was supported by the MRC as part of UK Research and Innovation (MRC grant MC_UP_1201/12 to J.T.P.Y.). J.J.R. is supported by a Herchel Smith Scholarship.

## Author contributions

J.J.R. conceptualized the study, purified proteins, performed all experiments, acquired and processed cryo-EM data, built and refined atomic models and wrote the original draft of the manuscript. J.T.P.Y. conceptualized and supervised the study, acquired funding and reviewed and edited the manuscript.

## Competing interests

The authors declare no competing interests.

## Additional information

**Extended data** is available for this paper at https://doi.org/10.1038/s41594-024-01370-y.

**Correspondence and requests for materials** should be addressed to Joseph T. P. Yeeles.

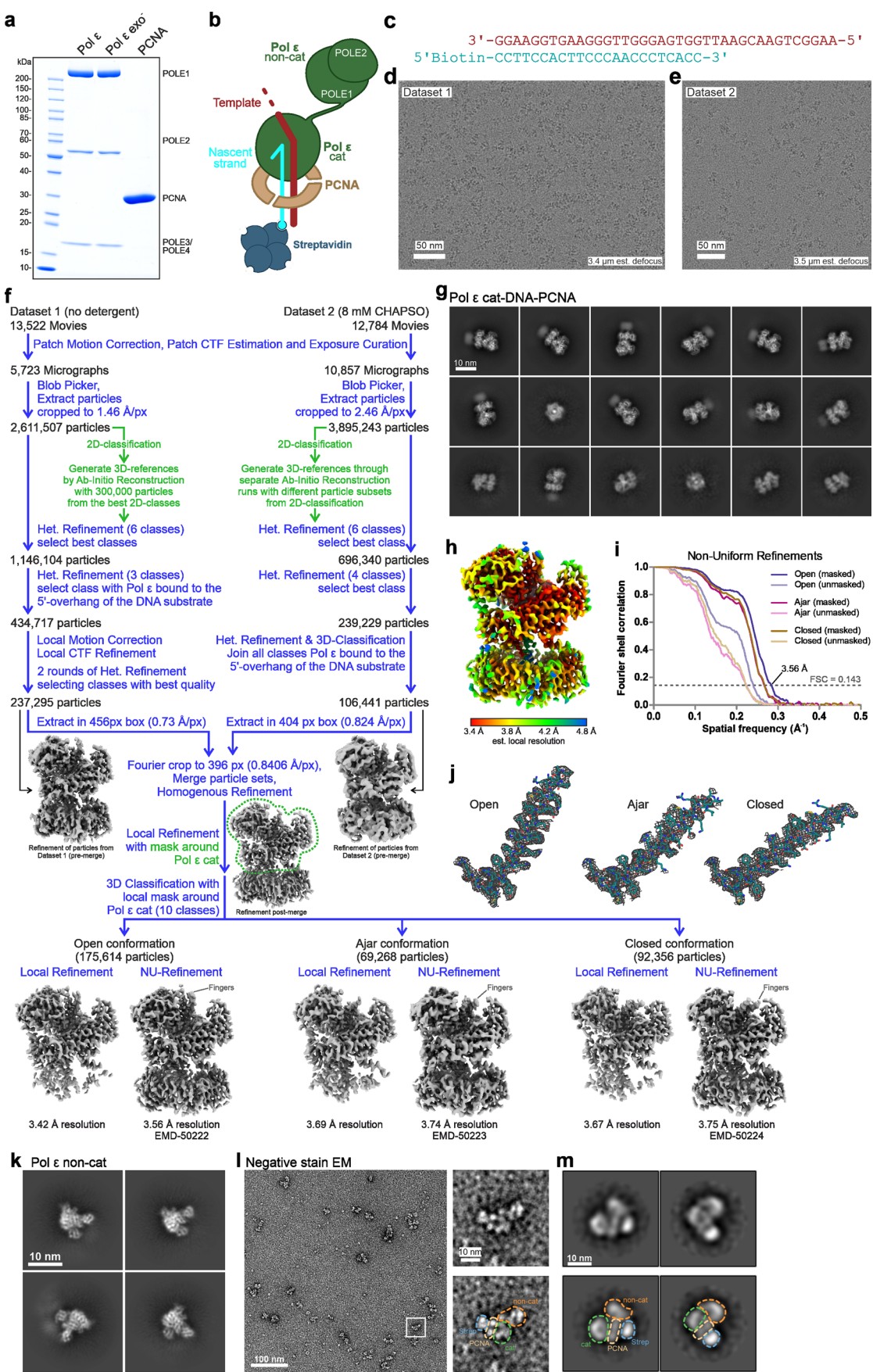

**c** 3'-GGAAGGTGAAGGGTTGGGAGTGGTTAAGCAAGTCGGAA-5'
5'Biotin-CCTTCCACTTCCCAACCCTCACC-3'

**Extended Data Fig. 1 | See next page for caption.**

**Extended Data Fig. 1 | Cryo-EM data processing of Pol ε-PCNA on matched DNA.** (**a**) Coomassie−stained SDS PAGE gels of purified Pol ε wt and exo⁻, and PCNA. (**b**) Schematic of Pol ε-DNA-PCNA reconstitution strategy on matched DNA. Streptavidin was added only in the sample for Dataset 1. (**c**) Template strand/nascent strand oligomer pair used as DNA substrate for reconstitution. Colours as in (b): template strand, red; nascent strand, cyan. (**d**, **e**) Example micrographs from datasets 1 (d) and 2 (e) after Patch Motion Correction and Patch CTF Estimation (estimated defocus is indicated). Scale bar, 50 nm. (**f**) Image processing workflow. Abbreviations: Het. Refinement, Heterogeneous Refinement; NU Refinement, Non-Uniform Refinement. (**g**) Averages of highest populated 2D classes generated from the final particle set in the open conformation using 2D-classification in CryoSPARC v4. Scale bar, 10 nm. (**h**) Refined 3D map in the open conformation coloured by local resolution calculated in CryoSPARC v4. (**i**) Masked and unmasked Fourier Shell Correlation (FSC) plots for the three final Non-Uniform Refinement maps shown in (f). Resolution is indicated at the FSC = 0.143 cut-off. (**j**) Cryo-EM density (mesh) for the second α-helix in the fingers domain for the indicated states. (**k**) Representative 2D-class averages from particles of dataset 1 that are aligned on the Pol ε noncatalytic domain. The scale bar indicates 10 nm. (**l**) Representative transmission electron micrograph taken on the same sample used for dataset 1 negatively stained with 2 % uranyl acetate on 3 nm carbon film and imaged at 26,000-fold magnification on a 120 kV FEI Tecnai Spirit Cryo using a Gatan Ultrascan 1000XP detector. A white square outlines one representative holoenzyme complex which is displayed on the right. (**m**) Representative 2D class averages from negative stain (top panel). Features of the Pol ε lobes, PCNA and Streptavidin are outlined in the bottom panel. The scale bar indicates 10 nm. Data shown in panels a, d, e and l are representative of experiments that were performed once.

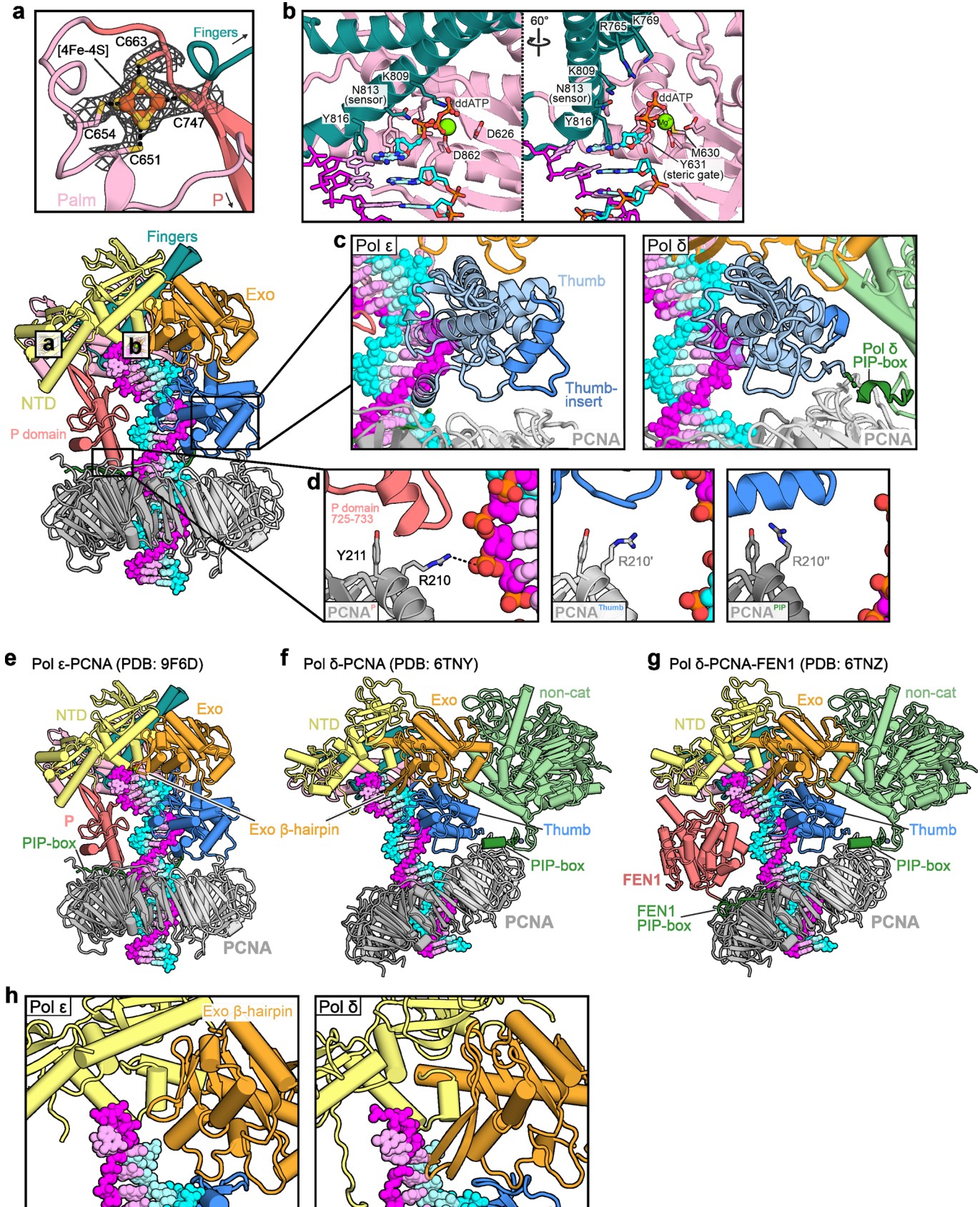

**Extended Data Fig. 2 | See next page for caption.**

**Extended Data Fig. 2 | Structure details of the Pol ε ternary complex with PCNA. (a)**. Cryo-EM density (shown as a mesh) around the iron-sulphur cluster (4Fe–4S) that is coordinated at the Pol ε CysX motif near the junction of P and Palm domains. (**b**) Two views of the polymerase active site of Pol ε in the open conformation. The incoming ddATP nucleotide and coordinating side chain residues are shown as sticks. (**c**) Comparison of the Thumb domains of Pol ε (left) and Pol δ (ref. 13) (right). The unique insert in Pol ε and the corresponding location in Pol δ are marked in light blue. The absence of this insert in Pol δ allows the PIP-box to interact with the PCNA protomer underneath the Thumb domain.

(**d**) Focused view on the contact site between the Pol ε P domain and the closest PCNA protomer (left). PCNA residues R210 and Y211 are shown as sticks. The dashed line indicates the electrostatic interaction formed between R210 and the DNA substrate. The conformation that these two side chains adopt in the other two PCNA protomers near adjacent to Thumb and PIP-box, respectively, are shown in the middle and right panels. (**e**–**g**) Ribbon models of Pol ε (e) and Pol δ (f, g)[13] in complex with PCNA, aligned by their catalytic domains. (**h**) Close−up view on the β-hairpin of the exonuclease domain (dark orange) for Pol ε (left) and Pol δ (ref. 13) (right).

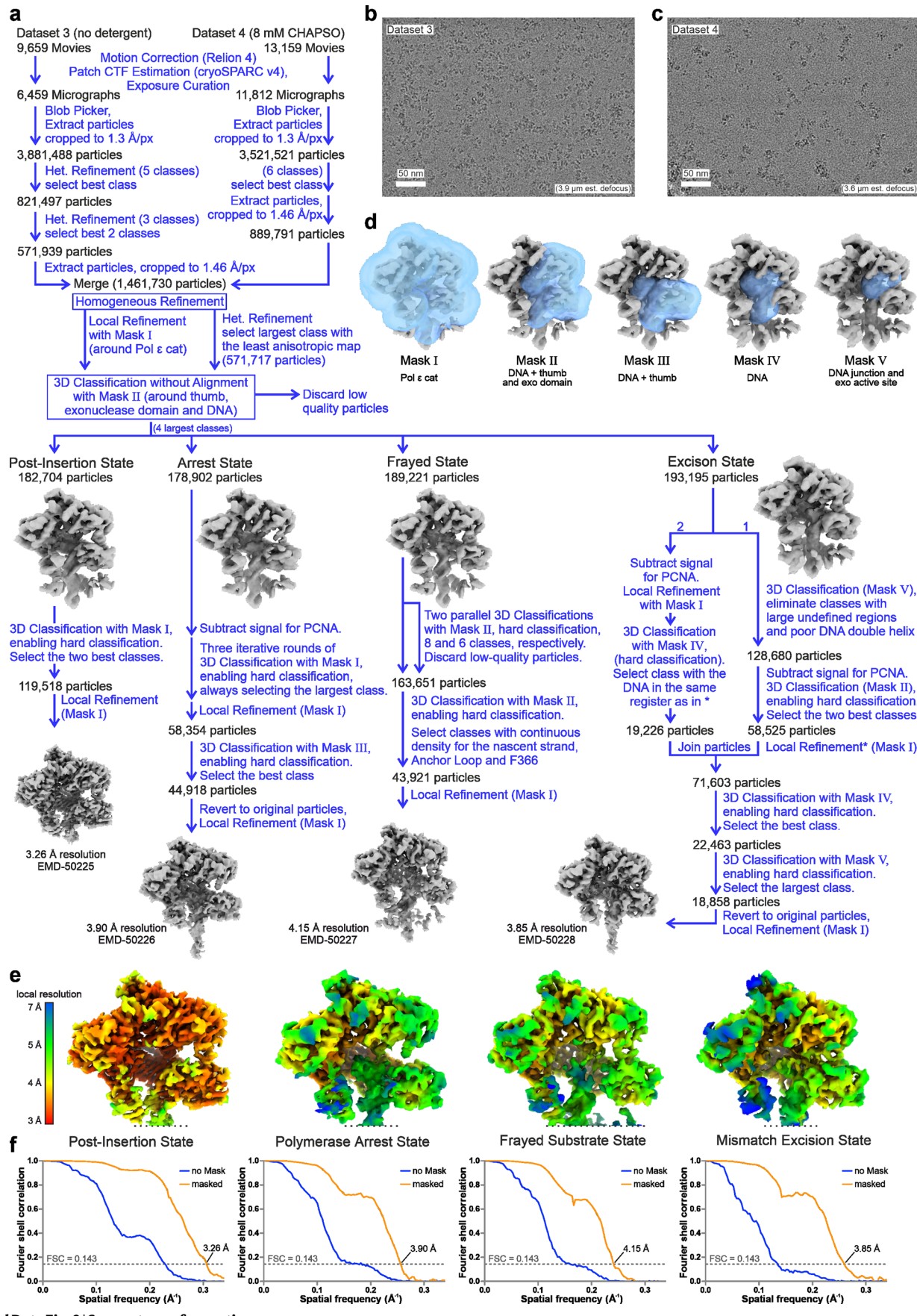

**Extended Data Fig. 3 | See next page for caption.**

**Extended Data Fig. 3 | Cryo-EM data processing of Pol ε-PCNA on mismatched DNA.** (**a**) Image processing workflow. Roman numerals refer to masks shown in (d). (**b**, **c**) Example micrographs from datasets 3 (**b**) and 4 (**c**) after Motion Correction and Patch CTF Estimation (estimated defocus is indicated). Scale bar, 50 nm. (**d**) 3D Masks used for 3D classification and focussed refinements are shown as blue transparent volumes around a reconstruction of Pol ε-PCNA (grey) for reference. (**e**) Refined 3D maps of the four described states coloured by local resolution calculated in CryoSPARC. (**f**) Masked and unmasked Fourier Shell Correlation (FSC) plots for the refined maps shown in (a, e). Resolution is indicated at the FSC = 0.143 cut-off. Data shown in panels b and c are representative of experiments that were performed once.

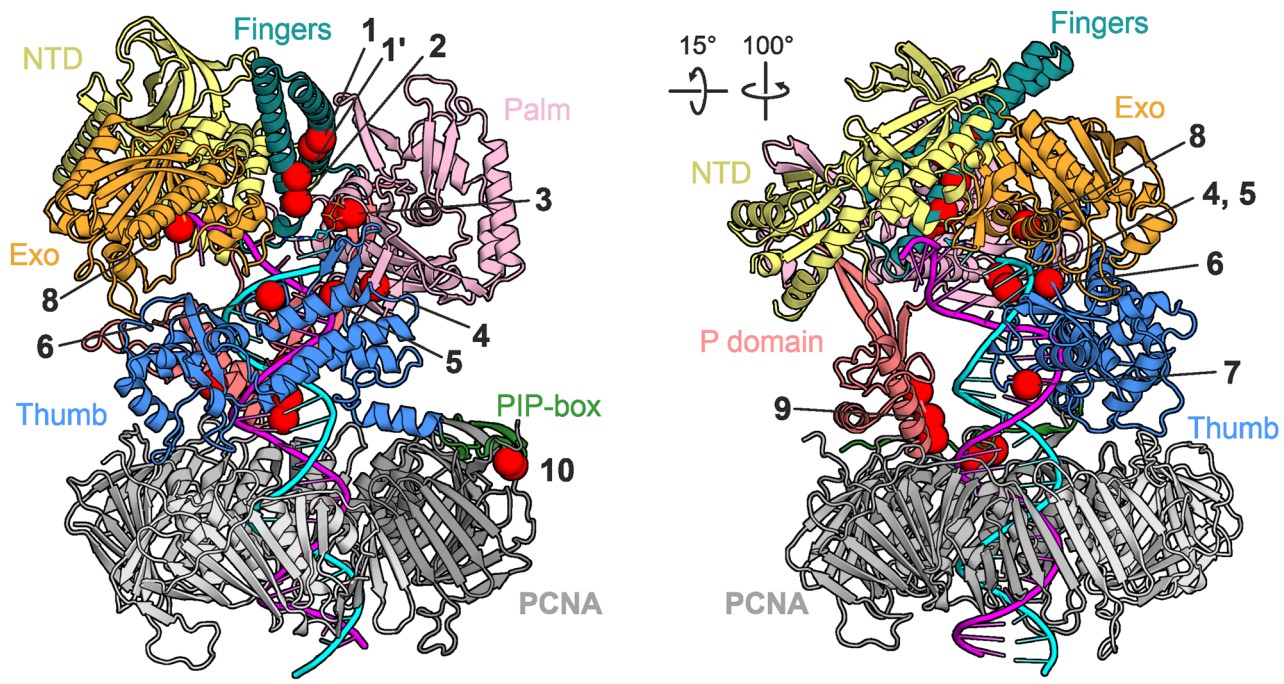

| | Location | Residues | Role in Polymerase Activity | |
|---|---|---|---|---|
| **1** | Fingers contacting the gamma-phosphate | R765 | Alanine exchange of K754 in Klenow or R482 in RB69 Pol results in impaired polymerase activity | Kaushik 1996, Yang 2002 |
| **1'** | Fingers contacting the alpha-phosphate | K809 | Alanine exchange of K758 in Klenow or K560 in RB69 Pol results in impaired polymerase activity | Kaushik 1996, Yang 2002 |
| **2** | Sensor | N813 | Discrimination against ribonucleotide incorporation | Parkash 2023 |
| **3** | Steric gate | M630, Y631 | Discrimination against ribonucleotide incorporation | Bonnin 1999, Pursell 2007 |
| **4** | Postinsertion site | K954 | Alanine exchange of K967 in *S. c.* Pol ε or K706 in RB96 Pol destabilises the nascent DNA 3'-terminus in the *pol* site | Ganai 2015, Zakharova 2004 |
| **5** | Palm-Thumb domain interface | E858 | Disruption (Y619F in RB69 Pol) impairs polymerase activity | Zakharova 2004 |
| **6** | Anchor Loop | R975 | *S. c.* R988 is critical for processive activity switching during proofreading | Ganai 2015 |
| **7** | Thumb domain tip | K1050 | K800 in RB69 Pol is implicated in polymerase translocation | Ren 2016 |
| **8** | Exo domain entry cavity | N363 | *S. c.* Pol ε N378K impairs partitioning to the exonuclease active site | Dahl 2022 |
| **9** | P domain | K732, K733, H735 | Alanine exchange in *S. c.* Pol ε abolishes processivity | Hogg 2015 |
| **10** | PIP box | L1186, F1187 | Alanine exchange of F1199 and F1200 in *S. c.* Pol ε results in loss of PCNA-dependent replication fork rates | Dua 2002, Aria 2019 |

**Extended Data Fig. 4 | Functional roles of Pol ε amino acid side chains involved in proofreading and activity switching.** Pol ε cat in complex with PCNA, substrate DNA and incoming nucleotide is shown as ribbons. The positions of selected amino acids are indicated by red spheres, numbered 1–10. The table lists the residues at the respective sites and their roles in the activity of Pol ε and/or other representative polymerases. References are: Kaushik 1996[61], Yang 2002[62], Parkash 2023[25], Bonnin 1999[26], Pursell 2007[63], Zakharova 2004[47], Ren 2016[64], Dahl 2022[48], Hogg 2014[17], Dua 2002[16], Aria 2019[52]. Abbreviation: *S. c.*, *Saccharomyces cerevisiae*.

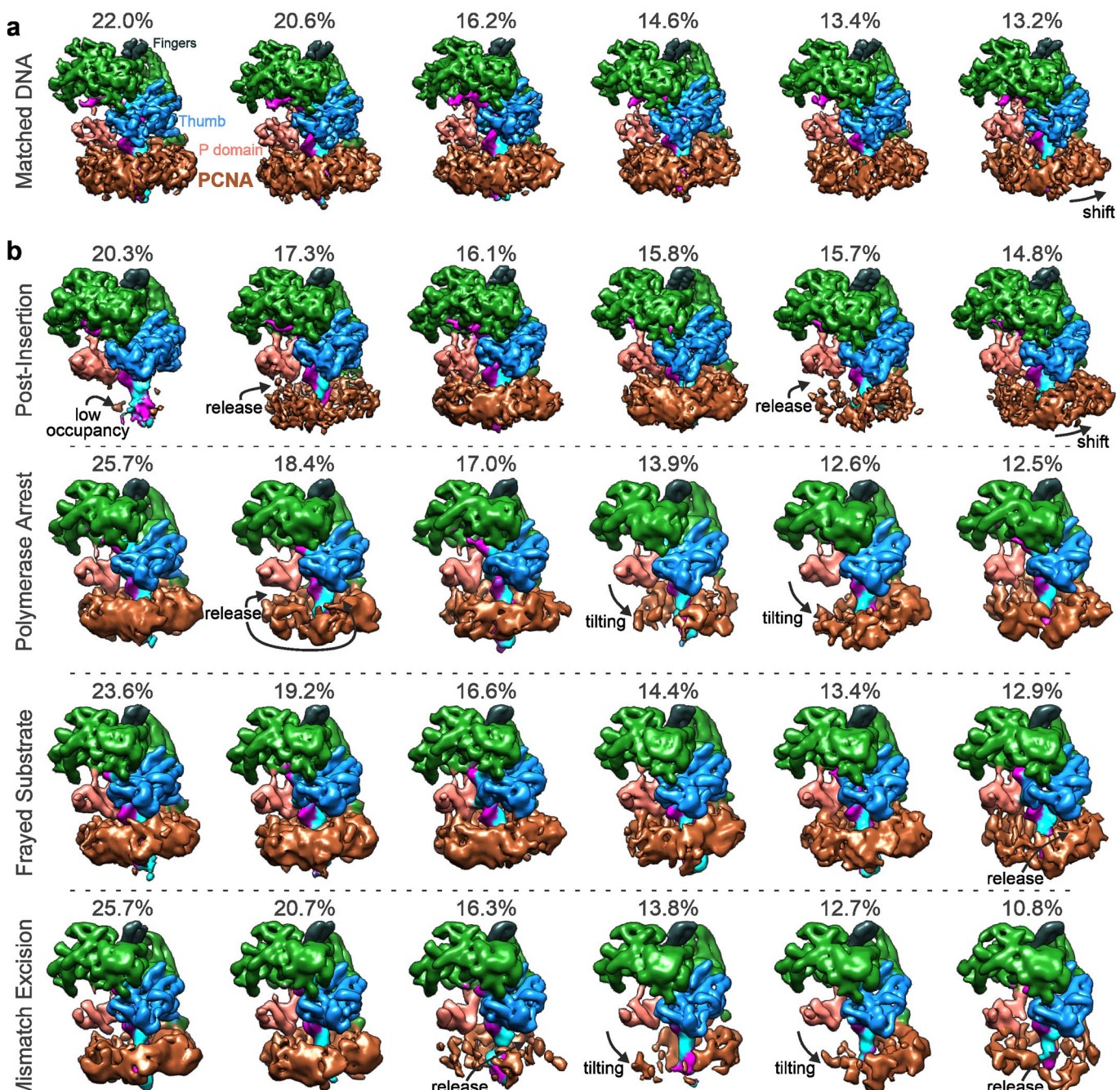

**Extended Data Fig. 5 | Variability analysis of PCNA in the different conformational states of Pol ε.** The final particle sets of Pol ε-PCNA on matched DNA in the open conformation (**a**) and from the four proofreading states (**b**) were 3D-classified in with a mask around PCNA. The displayed volumes are reconstructions from individual particle classes (particle populations are indicated in percentages of the input particles), filtered to 5 Å resolution. Large movements (shift and tilting) of PCNA and the release of contacts between PCNA and Thumb or P domains are indicated with arrows.

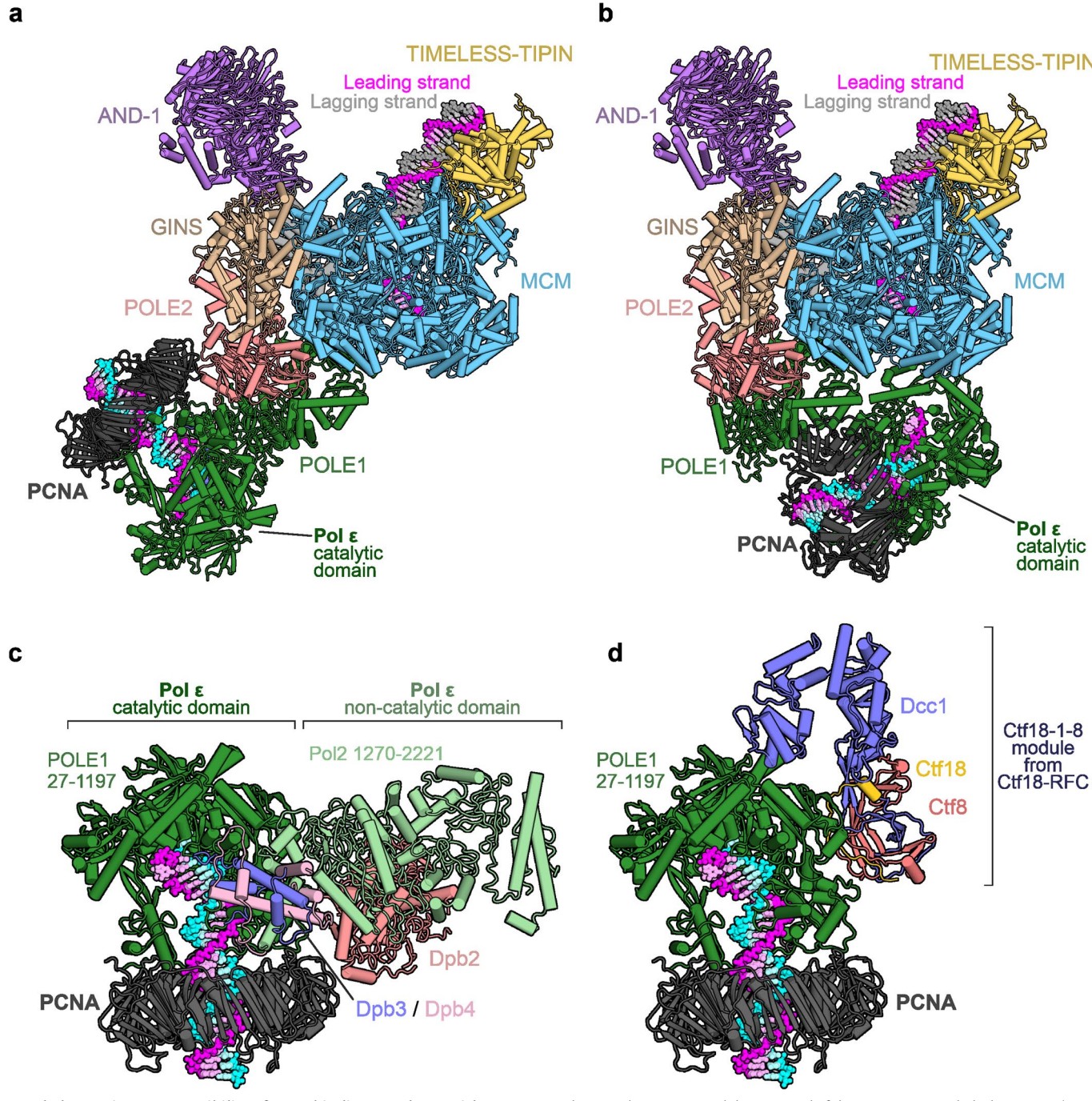

**Extended Data Fig. 6 | Compatibility of PCNA binding to Pol ε cat with other interactors of Pol ε.** (**a, b**) Pseudo-atomic models of the core human replisome (PDB: 7PFO, Jones 2021[36]) with the subcomplex of Pol ε cat-PCNA in two possible configurations. Pol ε cat-PCNA was placed either (a) based on the cryo-EM structure of the *S. cerevisiae* Pol ε holoenzyme (PDB: 6WJV, Yuan 2020[35]) or (b) based on reported cryo-EM density adjacent to the MCM channel exit in a structure containing *S. cerevisiae* CMG and Pol ε[53]. (**c**) Detailed view of

the pseudo-atomic model composed of the *S. cerevisiae* Pol ε holoenzyme (PDB: 6WJV, Yuan 2020[35]) and the PCNA interaction of the human ternary complex that we report here. Recruitment of PCNA is sterically compatible with the rigid conformation of Pol ε. (**d**) The interaction of the Pol ε ternary complex with PCNA is sterically compatible the recruitment of the Ctf18-1-8 module of the Ctf18-RFC clamp loader (PDB: 6S2F, Stokes 2020[54]).

# Reporting Summary

## Statistics

For all statistical analyses, confirm that the following items are present in the figure legend, table legend, main text, or Methods section.

| n/a | Confirmed | |
|---|---|---|
| ☒ | ☐ | The exact sample size (*n*) for each experimental group/condition, given as a discrete number and unit of measurement |
| ☒ | ☐ | A statement on whether measurements were taken from distinct samples or whether the same sample was measured repeatedly |
| ☒ | ☐ | The statistical test(s) used AND whether they are one- or two-sided<br>*Only common tests should be described solely by name; describe more complex techniques in the Methods section.* |
| ☒ | ☐ | A description of all covariates tested |
| ☒ | ☐ | A description of any assumptions or corrections, such as tests of normality and adjustment for multiple comparisons |
| ☐ | ☒ | A full description of the statistical parameters including central tendency (e.g. means) or other basic estimates (e.g. regression coefficient) AND variation (e.g. standard deviation) or associated estimates of uncertainty (e.g. confidence intervals) |
| ☒ | ☐ | For null hypothesis testing, the test statistic (e.g. *F*, *t*, *r*) with confidence intervals, effect sizes, degrees of freedom and *P* value noted<br>*Give P values as exact values whenever suitable.* |
| ☒ | ☐ | For Bayesian analysis, information on the choice of priors and Markov chain Monte Carlo settings |
| ☒ | ☐ | For hierarchical and complex designs, identification of the appropriate level for tests and full reporting of outcomes |
| ☒ | ☐ | Estimates of effect sizes (e.g. Cohen's *d*, Pearson's *r*), indicating how they were calculated |

*Our web collection on statistics for biologists contains articles on many of the points above.*

## Software and code

Policy information about availability of computer code

| Data collection | Gatan DigitalMicrograph and ThermoFisher EPU |
|---|---|
| Data analysis | RELION v4.0, cryoSPARC 4.3.0, UCSF ChimeraX v1.7.1, ISOLDE v1.7.1, Coot v0.9.3, Phenix v1.21, PyMOL v2.5.2 |

For manuscripts utilizing custom algorithms or software that are central to the research but not yet described in published literature, software must be made available to editors and reviewers. We strongly encourage code deposition in a community repository (e.g. GitHub). See the Nature Portfolio guidelines for submitting code & software for further information.

## Data

Policy information about availability of data

All manuscripts must include a data availability statement. This statement should provide the following information, where applicable:
- Accession codes, unique identifiers, or web links for publicly available datasets
- A description of any restrictions on data availability
- For clinical datasets or third party data, please ensure that the statement adheres to our policy

Structure coordinates have been deposited in the RCSB Protein Data Bank (https://www.rcsb.org/) and cryo-EM density maps used in model building have been deposited in the Electron Microscopy Data Bank (https://www.ebi.ac.uk/pdbe/emdb) under the following accession codes: 9F6D/EMD-50222 (Pol ε-PCNA on matched DNA, open Fingers conformation), 9F6E/EMD-50223 (Pol ε-PCNA on matched DNA, ajar Fingers), 9F6F/EMD-50224 (Pol ε-PCNA on matched DNA, closed Fingers), 9F6I/EMD-50225 (Pol ε-PCNA on mismatched DNA, Post-Insertion state), 9F6J/EMD-50226 (Pol ε-PCNA on mismatched DNA, Polymerase Arrest state), 9F6K/EMD-50227 (Pol ε-PCNA on mismatched DNA, Frayed state), 9F6L/EMD-50228 (Pol ε-PCNA on mismatched DNA, Mismatch Editing state).

# Field-specific reporting

Please select the one below that is the best fit for your research. If you are not sure, read the appropriate sections before making your selection.

☒ Life sciences      ☐ Behavioural & social sciences      ☐ Ecological, evolutionary & environmental sciences

For a reference copy of the document with all sections, see nature.com/documents/nr-reporting-summary-flat.pdf

# Life sciences study design

All studies must disclose on these points even when the disclosure is negative.

| | |
|---|---|
| Sample size | Our study does not include cohort/population based analysis or comparison and thus does not entail predetermination of sample size. Cryo-EM data were collected, as described in methods. In short, to yield high-resolution structures of human polymerase epsilon bound to DNA and PCNA, ~26.3k micrographs were collected. For high-resolution structures polymerase epsilon with mismatch-containing DNA of ~22.8k micrographs were collected. The numbers of micrographs and particles described in Methods and relevant Figures and Figure Legends were sufficient to generate maps of high resolution to allow model building and comparative analysis. |
| Data exclusions | During processing of cryo-EM data, poor quality micrographs/particles were excluded based on manual inspection and 2D/3D classification. |
| Replication | The different complexes identified in this study were visualised in multiple experiments (both negative stain and cryo-EM). In vitro formation of the described protein-nucleic acid complexes was found to be reproducible across at least three independent reconstitution reactions (determined by analytical size exclusion chromatography, negative stain EM and cryo-EM), all of which were successful. Cryo-EM specimen preparation (sample vitrification on grids) was reproducible across cryo-EM grids with the same sample as well as across at least two independently prepared samples for each complex type, all of which were successful. |
| Randomization | The resolution of the cryo-EM reconstructions was determined via Fourier shell correlations which were calculated using independent halves of the complete datasets, into which the component particles were segregated randomly. |
| Blinding | Blinding is not relevant for a single particle electron microscopy study such as this since there is no identifiable risk of bias. |

# Behavioural & social sciences study design

All studies must disclose on these points even when the disclosure is negative.

| | |
|---|---|
| Study description | Briefly describe the study type including whether data are quantitative, qualitative, or mixed-methods (e.g. qualitative cross-sectional, quantitative experimental, mixed-methods case study). |
| Research sample | State the research sample (e.g. Harvard university undergraduates, villagers in rural India) and provide relevant demographic information (e.g. age, sex) and indicate whether the sample is representative. Provide a rationale for the study sample chosen. For studies involving existing datasets, please describe the dataset and source. |
| Sampling strategy | Describe the sampling procedure (e.g. random, snowball, stratified, convenience). Describe the statistical methods that were used to predetermine sample size OR if no sample-size calculation was performed, describe how sample sizes were chosen and provide a rationale for why these sample sizes are sufficient. For qualitative data, please indicate whether data saturation was considered, and what criteria were used to decide that no further sampling was needed. |
| Data collection | Provide details about the data collection procedure, including the instruments or devices used to record the data (e.g. pen and paper, computer, eye tracker, video or audio equipment) whether anyone was present besides the participant(s) and the researcher, and whether the researcher was blind to experimental condition and/or the study hypothesis during data collection. |
| Timing | Indicate the start and stop dates of data collection. If there is a gap between collection periods, state the dates for each sample cohort. |
| Data exclusions | If no data were excluded from the analyses, state so OR if data were excluded, provide the exact number of exclusions and the rationale behind them, indicating whether exclusion criteria were pre-established. |
| Non-participation | State how many participants dropped out/declined participation and the reason(s) given OR provide response rate OR state that no participants dropped out/declined participation. |
| Randomization | If participants were not allocated into experimental groups, state so OR describe how participants were allocated to groups, and if allocation was not random, describe how covariates were controlled. |

# Ecological, evolutionary & environmental sciences study design

All studies must disclose on these points even when the disclosure is negative.

| | |
|---|---|
| Study description | *Briefly describe the study. For quantitative data include treatment factors and interactions, design structure (e.g. factorial, nested, hierarchical), nature and number of experimental units and replicates.* |
| Research sample | *Describe the research sample (e.g. a group of tagged Passer domesticus, all Stenocereus thurberi within Organ Pipe Cactus National Monument), and provide a rationale for the sample choice. When relevant, describe the organism taxa, source, sex, age range and any manipulations. State what population the sample is meant to represent when applicable. For studies involving existing datasets, describe the data and its source.* |
| Sampling strategy | *Note the sampling procedure. Describe the statistical methods that were used to predetermine sample size OR if no sample-size calculation was performed, describe how sample sizes were chosen and provide a rationale for why these sample sizes are sufficient.* |
| Data collection | *Describe the data collection procedure, including who recorded the data and how.* |
| Timing and spatial scale | *Indicate the start and stop dates of data collection, noting the frequency and periodicity of sampling and providing a rationale for these choices. If there is a gap between collection periods, state the dates for each sample cohort. Specify the spatial scale from which the data are taken* |
| Data exclusions | *If no data were excluded from the analyses, state so OR if data were excluded, describe the exclusions and the rationale behind them, indicating whether exclusion criteria were pre-established.* |
| Reproducibility | *Describe the measures taken to verify the reproducibility of experimental findings. For each experiment, note whether any attempts to repeat the experiment failed OR state that all attempts to repeat the experiment were successful.* |
| Randomization | *Describe how samples/organisms/participants were allocated into groups. If allocation was not random, describe how covariates were controlled. If this is not relevant to your study, explain why.* |
| Blinding | *Describe the extent of blinding used during data acquisition and analysis. If blinding was not possible, describe why OR explain why blinding was not relevant to your study.* |

Did the study involve field work? ☐ Yes ☐ No

# Field work, collection and transport

| | |
|---|---|
| Field conditions | *Describe the study conditions for field work, providing relevant parameters (e.g. temperature, rainfall).* |
| Location | *State the location of the sampling or experiment, providing relevant parameters (e.g. latitude and longitude, elevation, water depth).* |
| Access & import/export | *Describe the efforts you have made to access habitats and to collect and import/export your samples in a responsible manner and in compliance with local, national and international laws, noting any permits that were obtained (give the name of the issuing authority, the date of issue, and any identifying information).* |
| Disturbance | *Describe any disturbance caused by the study and how it was minimized.* |

# Reporting for specific materials, systems and methods

We require information from authors about some types of materials, experimental systems and methods used in many studies. Here, indicate whether each material, system or method listed is relevant to your study. If you are not sure if a list item applies to your research, read the appropriate section before selecting a response.

## Materials & experimental systems

| n/a | Involved in the study |
|---|---|
| ☒ | ☐ Antibodies |
| ☐ | ☒ Eukaryotic cell lines |
| ☒ | ☐ Palaeontology and archaeology |
| ☒ | ☐ Animals and other organisms |
| ☒ | ☐ Human research participants |
| ☒ | ☐ Clinical data |
| ☒ | ☐ Dual use research of concern |

## Methods

| n/a | Involved in the study |
|---|---|
| ☒ | ☐ ChIP-seq |
| ☒ | ☐ Flow cytometry |
| ☒ | ☐ MRI-based neuroimaging |

# Antibodies

Antibodies used

*Describe all antibodies used in the study; as applicable, provide supplier name, catalog number, clone name, and lot number.*

Validation

*Describe the validation of each primary antibody for the species and application, noting any validation statements on the manufacturer's website, relevant citations, antibody profiles in online databases, or data provided in the manuscript.*

# Eukaryotic cell lines

Policy information about cell lines

Cell line source(s)

SF9 cells were obtained from OXFORD EXPRESSION TECHNOLOGIES, LTD, Cat No. 600100. High-5 cells (bti-tn-5b1-4) were obtained from Thermo-Scientific Cat no. B85502.

Authentication

None of the cell lines were authenticated.

Mycoplasma contamination

Sf9 and High-5 cells were tested negative for mycoplasma contamination.

Commonly misidentified lines
(See ICLAC register)

No commonly misidentified cell lines were used in this study.

# Palaeontology and Archaeology

Specimen provenance

*Provide provenance information for specimens and describe permits that were obtained for the work (including the name of the issuing authority, the date of issue, and any identifying information). Permits should encompass collection and, where applicable, export.*

Specimen deposition

*Indicate where the specimens have been deposited to permit free access by other researchers.*

Dating methods

*If new dates are provided, describe how they were obtained (e.g. collection, storage, sample pretreatment and measurement), where they were obtained (i.e. lab name), the calibration program and the protocol for quality assurance OR state that no new dates are provided.*

☐ Tick this box to confirm that the raw and calibrated dates are available in the paper or in Supplementary Information.

Ethics oversight

*Identify the organization(s) that approved or provided guidance on the study protocol, OR state that no ethical approval or guidance was required and explain why not.*

Note that full information on the approval of the study protocol must also be provided in the manuscript.

# Animals and other organisms

Policy information about studies involving animals; ARRIVE guidelines recommended for reporting animal research

Laboratory animals

*For laboratory animals, report species, strain, sex and age OR state that the study did not involve laboratory animals.*

Wild animals

*Provide details on animals observed in or captured in the field; report species, sex and age where possible. Describe how animals were caught and transported and what happened to captive animals after the study (if killed, explain why and describe method; if released, say where and when) OR state that the study did not involve wild animals.*

Field-collected samples

*For laboratory work with field-collected samples, describe all relevant parameters such as housing, maintenance, temperature, photoperiod and end-of-experiment protocol OR state that the study did not involve samples collected from the field.*

Ethics oversight

*Identify the organization(s) that approved or provided guidance on the study protocol, OR state that no ethical approval or guidance was required and explain why not.*

Note that full information on the approval of the study protocol must also be provided in the manuscript.

# Human research participants

Policy information about studies involving human research participants

Population characteristics

*Describe the covariate-relevant population characteristics of the human research participants (e.g. age, gender, genotypic information, past and current diagnosis and treatment categories). If you filled out the behavioural & social sciences study design questions and have nothing to add here, write "See above."*

Recruitment

*Describe how participants were recruited. Outline any potential self-selection bias or other biases that may be present and how these are likely to impact results.*

Ethics oversight

*Identify the organization(s) that approved the study protocol.*

Note that full information on the approval of the study protocol must also be provided in the manuscript.

# Clinical data

Policy information about clinical studies
All manuscripts should comply with the ICMJE guidelines for publication of clinical research and a completed CONSORT checklist must be included with all submissions.

Clinical trial registration | *Provide the trial registration number from ClinicalTrials.gov or an equivalent agency.*

Study protocol | *Note where the full trial protocol can be accessed OR if not available, explain why.*

Data collection | *Describe the settings and locales of data collection, noting the time periods of recruitment and data collection.*

Outcomes | *Describe how you pre-defined primary and secondary outcome measures and how you assessed these measures.*

# Dual use research of concern

Policy information about dual use research of concern

## Hazards

Could the accidental, deliberate or reckless misuse of agents or technologies generated in the work, or the application of information presented in the manuscript, pose a threat to:

No | Yes
☐ | ☐ Public health
☐ | ☐ National security
☐ | ☐ Crops and/or livestock
☐ | ☐ Ecosystems
☐ | ☐ Any other significant area

## Experiments of concern

Does the work involve any of these experiments of concern:

No | Yes
☐ | ☐ Demonstrate how to render a vaccine ineffective
☐ | ☐ Confer resistance to therapeutically useful antibiotics or antiviral agents
☐ | ☐ Enhance the virulence of a pathogen or render a nonpathogen virulent
☐ | ☐ Increase transmissibility of a pathogen
☐ | ☐ Alter the host range of a pathogen
☐ | ☐ Enable evasion of diagnostic/detection modalities
☐ | ☐ Enable the weaponization of a biological agent or toxin
☐ | ☐ Any other potentially harmful combination of experiments and agents

# ChIP-seq

## Data deposition

☐ Confirm that both raw and final processed data have been deposited in a public database such as GEO.

☐ Confirm that you have deposited or provided access to graph files (e.g. BED files) for the called peaks.

Data access links
*May remain private before publication.* | *For "Initial submission" or "Revised version" documents, provide reviewer access links.  For your "Final submission" document, provide a link to the deposited data.*

Files in database submission | *Provide a list of all files available in the database submission.*

Genome browser session
(e.g. UCSC) | *Provide a link to an anonymized genome browser session for "Initial submission" and "Revised version" documents only, to enable peer review.  Write "no longer applicable" for "Final submission" documents.*

## Methodology

Replicates | *Describe the experimental replicates, specifying number, type and replicate agreement.*

Sequencing depth | *Describe the sequencing depth for each experiment, providing the total number of reads, uniquely mapped reads, length of reads and*

| Sequencing depth | *whether they were paired- or single-end.* |
|---|---|
| Antibodies | *Describe the antibodies used for the ChIP-seq experiments; as applicable, provide supplier name, catalog number, clone name, and lot number.* |
| Peak calling parameters | *Specify the command line program and parameters used for read mapping and peak calling, including the ChIP, control and index files used.* |
| Data quality | *Describe the methods used to ensure data quality in full detail, including how many peaks are at FDR 5% and above 5-fold enrichment.* |
| Software | *Describe the software used to collect and analyze the ChIP-seq data. For custom code that has been deposited into a community repository, provide accession details.* |

## Flow Cytometry

### Plots

Confirm that:

☐ The axis labels state the marker and fluorochrome used (e.g. CD4-FITC).

☐ The axis scales are clearly visible. Include numbers along axes only for bottom left plot of group (a 'group' is an analysis of identical markers).

☐ All plots are contour plots with outliers or pseudocolor plots.

☐ A numerical value for number of cells or percentage (with statistics) is provided.

### Methodology

| Sample preparation | *Describe the sample preparation, detailing the biological source of the cells and any tissue processing steps used.* |
|---|---|
| Instrument | *Identify the instrument used for data collection, specifying make and model number.* |
| Software | *Describe the software used to collect and analyze the flow cytometry data. For custom code that has been deposited into a community repository, provide accession details.* |
| Cell population abundance | *Describe the abundance of the relevant cell populations within post-sort fractions, providing details on the purity of the samples and how it was determined.* |
| Gating strategy | *Describe the gating strategy used for all relevant experiments, specifying the preliminary FSC/SSC gates of the starting cell population, indicating where boundaries between "positive" and "negative" staining cell populations are defined.* |

☐ Tick this box to confirm that a figure exemplifying the gating strategy is provided in the Supplementary Information.

## Magnetic resonance imaging

### Experimental design

| Design type | *Indicate task or resting state; event-related or block design.* |
|---|---|
| Design specifications | *Specify the number of blocks, trials or experimental units per session and/or subject, and specify the length of each trial or block (if trials are blocked) and interval between trials.* |
| Behavioral performance measures | *State number and/or type of variables recorded (e.g. correct button press, response time) and what statistics were used to establish that the subjects were performing the task as expected (e.g. mean, range, and/or standard deviation across subjects).* |

### Acquisition

| Imaging type(s) | *Specify: functional, structural, diffusion, perfusion.* |
|---|---|
| Field strength | *Specify in Tesla* |
| Sequence & imaging parameters | *Specify the pulse sequence type (gradient echo, spin echo, etc.), imaging type (EPI, spiral, etc.), field of view, matrix size, slice thickness, orientation and TE/TR/flip angle.* |
| Area of acquisition | *State whether a whole brain scan was used OR define the area of acquisition, describing how the region was determined.* |

Diffusion MRI   ☐ Used   ☐ Not used

## Preprocessing

**Preprocessing software**
*Provide detail on software version and revision number and on specific parameters (model/functions, brain extraction, segmentation, smoothing kernel size, etc.).*

**Normalization**
*If data were normalized/standardized, describe the approach(es): specify linear or non-linear and define image types used for transformation OR indicate that data were not normalized and explain rationale for lack of normalization.*

**Normalization template**
*Describe the template used for normalization/transformation, specifying subject space or group standardized space (e.g. original Talairach, MNI305, ICBM152) OR indicate that the data were not normalized.*

**Noise and artifact removal**
*Describe your procedure(s) for artifact and structured noise removal, specifying motion parameters, tissue signals and physiological signals (heart rate, respiration).*

**Volume censoring**
*Define your software and/or method and criteria for volume censoring, and state the extent of such censoring.*

## Statistical modeling & inference

**Model type and settings**
*Specify type (mass univariate, multivariate, RSA, predictive, etc.) and describe essential details of the model at the first and second levels (e.g. fixed, random or mixed effects; drift or auto-correlation).*

**Effect(s) tested**
*Define precise effect in terms of the task or stimulus conditions instead of psychological concepts and indicate whether ANOVA or factorial designs were used.*

**Specify type of analysis:** ☐ Whole brain ☐ ROI-based ☐ Both

**Statistic type for inference**
(See Eklund et al. 2016)
*Specify voxel-wise or cluster-wise and report all relevant parameters for cluster-wise methods.*

**Correction**
*Describe the type of correction and how it is obtained for multiple comparisons (e.g. FWE, FDR, permutation or Monte Carlo).*

## Models & analysis

| n/a | Involved in the study |
|-----|----------------------|
| ☐ | ☐ Functional and/or effective connectivity |
| ☐ | ☐ Graph analysis |
| ☐ | ☐ Multivariate modeling or predictive analysis |

**Functional and/or effective connectivity**
*Report the measures of dependence used and the model details (e.g. Pearson correlation, partial correlation, mutual information).*

**Graph analysis**
*Report the dependent variable and connectivity measure, specifying weighted graph or binarized graph, subject- or group-level, and the global and/or node summaries used (e.g. clustering coefficient, efficiency, etc.).*

**Multivariate modeling and predictive analysis**
*Specify independent variables, features extraction and dimension reduction, model, training and evaluation metrics.*

