## [Peer Review File · Nature Structural & Molecular Biology]

Peer Review Information

Manuscript Title: Structural basis for processive daughter strand synthesis and proofreading by the human leading-strand DNA polymerase Pol ϵ

Corresponding author name(s): Joseph Yeeles

Reviewer Comments & Decisions:

Decision Letter, initial version:

Message: 22nd Apr 2024

Dear Dr. Yeeles,

Thank you again for submitting your manuscript "Structural basis for processive daughter strand synthesis and proofreading by the human leading-strand polymerase Pol ϵ ". We now have comments (below) from the 3 reviewers who evaluated your paper. In light of those reports, we remain interested in your study and would like to see your response to the comments of the referees, in the form of a revised manuscript.

You will see that all reviewers are appreciative of the mechanistic and structural findings, with reviewers #1 and #3 posing a single mechanistic question each. However, reviewer #2 raises a few important questions which necessitate convincing addressing both experimentally and textually in a revised manuscript. More specifically, we note the request for additional biochemical/mutational analyses with respect to Polymerase epsilon interactions with multiple PCNA protomers and the need for further support for side chain modelling of certain residues in the 'frayed substrate' state. Additionally, we ask you to please address all the questions and provide the requested clarifications, additional discussion and potentially missing information in accordance to the guidance of the experts.

Please be sure to respond to all concerns of the referees in full in a point-by-point response and highlight all changes in the revised manuscript text file. If you have comments that are intended for editors only, please include those in a separate cover letter.

We expect to see your revised manuscript within 6 weeks. If you cannot send it within this time, please contact us to discuss an extension; we would still consider your revision, provided that no similar work has been accepted for publication at NSMB or published elsewhere.

Reporting Summary:

When submitting the revised version of your manuscript, please pay close attention to our [href="https://www.nature.com/nature-portfolio/editorial-policies/image-integrity">Digital Image Integrity Guidelines](https://www.nature.com/nature-portfolio/editorial-policies/image-integrity). and to the following points below:

Data availability: this journal strongly supports public availability of data. All data used in accepted papers should be available via a public data repository, or alternatively, as Supplementary Information. If data can only be shared on request, please explain why in

your Data Availability Statement, and also in the correspondence with your editor. Please note that for some data types, deposition in a public repository is mandatory - more information on our data deposition policies and available repositories can be found below: <https://www.nature.com/nature-research/editorial-policies/reporting-standards#availability-of-data>

Nature Structural & Molecular Biology is committed to improving transparency in authorship. As part of our efforts in this direction, we are now requesting that all authors identified as 'corresponding author' on published papers create and link their Open Researcher and Contributor Identifier (ORCID) with their account on the Manuscript Tracking System (MTS), prior to acceptance. This applies to primary research papers only. ORCID helps the scientific community achieve unambiguous attribution of all scholarly contributions. You can create and link your ORCID from the home page of the MTS by clicking on 'Modify my Springer Nature account'. For more information please visit please visit www.springernature.com/orcid.

[Redacted]

Sincerely,

Dimitris Typas
Associate Editor
Nature Structural & Molecular Biology
ORCID: 0000-0002-8737-1319

Reviewers' Comments:

Reviewer #1:

Remarks to the Author:

In this paper the authors present the first structure of human Pol epsilon in complex with various DNA substrates that reveals snapshots from the dynamic process when Pol epsilon builds and correct its own errors (proofreading). The paper is limited to structural studies and relies on earlier biochemical studies to explain some of the features in this novel structure. The authors not only describe the interaction between Pol epsilon and PCNA, they also in detail describe how Pol epsilon achieves high fidelity when building new DNA. There is no need to carry out additional biochemistry since the features in this structure are corroborated by previously published papers. In my opinion this will be a seminal paper and a highly cited paper because of the information in the presented structures and today there is a great interest in variants of Pol epsilon that are drivers in the development of cancer.

I have only one minor question:

Can you elaborate on why Pol epsilon has a low affinity for PCNA in solution/off DNA? You suggest that the key interaction is the one with the PIP-box and the same PIP-box allows Pol delta and many other proteins to interact with PCNA in solution/off DNA.

Reviewer #2:

Remarks to the Author:

In this study, Roske and Yeeles report cryo-EM structures of the catalytic domain of human DNA polymerase epsilon (Pol ϵ) bound to the sliding clamp PCNA, along with various DNA substrates. Specifically, they resolved structures that represent the holoenzyme during processive elongation on a matched substrate, as well as the end-point states (referred to in the paper as 'post-insertion' and 'excision') in the pathway of transferring a mismatched substrate from the polymerase to the exonuclease site of Pol ϵ for proofreading. Additionally, two novel states (referred to as 'arrest' and 'frayed substrate') are presented, which the authors assigned as intermediates between the post-insertion and excision states.

In the cryo-EM work, constructs of Pol ϵ that included the entire POLE1 subunit along with auxiliary subunits were used. However, only the catalytic domain of POLE1 (POLE1_CAT) was resolved in the maps, suggesting that POLE1_CAT is flexibly connected to the non-catalytic domain. The structure of POLE1_CAT closely resembles that of the yeast homologue (Hogg et al, NSMB, 2014; Jain et al, PLoS one, 2014).

The general architecture of the POLE1_CAT-PCNA-DNA complexes mirrors those of other DNA polymerases bound to PCNA that have been previously published (Zheng et al, PNAS, 2020; Lancey et al, Nat Comm, 2020; Lancey et al, Nat Comm, 2021; Madru et al, Nat Comm, 2020). In these complexes, the polymerase sits atop PCNA, with DNA exiting the catalytic domain and threading through the PCNA ring. In the Pol ϵ complex, the primary interaction with PCNA involves a typical, well conserved PIP-box at the POLE1_CAT C-terminus binding to one PCNA protomer, along with additional, minor contacts between the P and Thumb domains of the polymerase and the other two PCNA protomers.

Moreover, the structure of the post-insertion state of the complex with a mismatched

substrate resembles the analogous complex of human B-family Pol α -primase (Baranovskiy et al, PNAS, 2022). This similarity includes impaired stacking between the mismatch and the nascent base pair and a suboptimal angle between the 3'-OH and the α -phosphate, important for catalysis. The excision state of the complex closely mirrors that of the B-family phage RB69 polymerase (Shamoo and Steitz, Cell, 1999).

The study introduces novel 'arrest' and 'frayed' states, proposed to link the post-insertion and excision states. Collectively, the authors propose a mechanism for transferring the mismatch to the polymerase and exonuclease sites, involving the expulsion of the DNA substrate from the active site by the thumb domain, fraying of two bases at the 3' end of the nascent strand, and the final 3'-end transfer to the exonuclease site. Incidentally, a "frayed primer-template" model for proof reading has been already proposed for human A-family DNA polymerase γ , based on multiple structures of intermediates between polymerization and editing states (Park et al, NSMB, 2023; Buchel et al., Nat Comm, 2023).

While the study is technically sound, because of previous structural work revealing key aspects of DNA polymerases (particularly regarding clamp-enhanced processivity and proofreading), the authors should explain more clearly the novelty in their work.

Major points:

- What are the structural determinants that underlie the differing functional dependencies of Pol ϵ and Pol δ on PCNA (beyond the inability of Pol ϵ -PCNA to form toolbelts with other factors)? Previous studies have demonstrated that while Pol δ is entirely dependent on PCNA for its function in DNA replication, Pol ϵ is not (see, for instance, Georgescu et al, NSMB, 2014; Lee, Hurwitz et al, JBC, 1991). The authors should further elaborate on the unique mechanisms by which PCNA enhances the processivity of Pol ϵ
- I found it surprising that Pol ϵ shows interactions with multiple PCNA protomers, indicating a high-affinity interaction, which contradicts previous reports (for instance, see Chilkova et al, NAR, 2007). Without mutational or functional analysis, the significance of the novel contact points between the P and thumb domains with PCNA on Pol ϵ function remains unclear. The authors should clarify why streptavidin was conjugated to one end of the DNA substrates. Was it used to prevent PCNA from sliding off the substrate and to enhance its affinity for Pol ϵ ?
- The observed flexibility between POLE1_CAT, the POLE1 dead domain, and ancillary subunits in these structures contrasts with a previous structural report on the yeast homolog, which depicted a rigid complex (Yuan, Nat Comm, 2020). The authors should discuss this discrepancy and evaluate whether the interactions of POLE1_CAT with PCNA could also occur in a rigid Pol ϵ complex.
- The resolution of the 'frayed substrate' state map at 4.14Å poses challenges for side chain modeling. The map lacks density for the side chains of residues K732, K733, and R728 in Pol ϵ , yet the authors suggest these residues help tightly bind the DNA to the polymerase. Additionally, the weak density for the 3'-end of the nascent strand and its apparent looseness raise questions about the interactions responsible for melting from the template strand.
- The authors should clarify the structural determinants that prevent the fingers domain from closing in the post-insertion state, despite the presence of a complementary nucleotide in the active site.
- For enhanced clarity, it would be beneficial if the authors could produce morphing movies that show transitions between the four proofreading states, highlighting the

evolution of key protein-DNA interactions

Minor points:

- The authors say that "Pol δ binds PCNA via a highly conserved PCNA-interacting peptide (PIP-box) motif in the large polymerase subunit p125 (or POLD1)". However, the PIP-box is not highly conserved and is atypical (Zheng et al, PNAS, 2020; Lancey et al, Nat Comm, 2020). In addition, other interactions contribute in binding the Pol to PCNA.
- What are the axes of rotation shown in Fig. 2b and Fig. 5b?

Reviewer #3:

Remarks to the Author:

This manuscript presents cryo-EM structures of human DNA polymerase epsilon in the process of transitioning a T•C mismatch from the polymerase active site to the exonuclease active site to remove the incorrect terminal nucleotide from single-stranded DNA. In my opinion, the study is beautifully performed and described, and it is worthy of publication in NSMB. The new structures imply that proofreading by this enzyme can occur without enzyme dissociation, a point that will be of great interest to experts interested in the mechanisms that determine accurate replication. The study is highly relevant to the emerging data published by others suggesting that proofreading by Pol epsilon, which is hypothesized to primarily replicate the leading DNA strand during replication, differs somewhat from DNA polymerase delta. (This could possibly be discussed in greater detail, either here or at some point in the future.) In addition, the study should be of great interest to scientists who study the origins of human diseases, and for several reasons, to evolutionary biologists. I congratulate these two authors on a fine study. The observations made here will form the baseline for much future work.

On what basis was a T•C mismatch chosen for this study? Are there any issues worth considering given that it is but one mismatch among billions that are potentially relevant to replication of the human nuclear genome?

Author Rebuttal to Initial comments

Dear editors and reviewers,

Thank you very much for the time you have spent assessing our manuscript. We are very pleased with the positive responses to our work. We are confident that we have satisfactorily addressed the specific reviewer comments. Details of how we have done so are outlined in our point-by-point response that is provided below. All changes to the main manuscript are highlighted in green.

Reviewer #1:

Remarks to the Author:

In this paper the authors present the first structure of human Pol epsilon in complex with various DNA substrates that reveals snapshots from the dynamic process when Pol epsilon builds and corrects its own errors (proofreading). The paper is limited to structural studies and relies on earlier biochemical studies to explain some of the features in this novel structure. The authors not only describe the interaction between Pol epsilon and PCNA, they also in detail describe how Pol epsilon achieves high fidelity when building new DNA. There is no need to carry out additional biochemistry since the features in this structure are corroborated by previously published papers. In my opinion this will be a seminal paper and a highly cited paper because of the information in the presented structures and today there is a great interest in variants of Pol epsilon that are drivers in the development of cancer.

We thank the reviewer for their enthusiastic support of our manuscript.

I have only one minor question:

Can you elaborate on why Pol epsilon has a low affinity for PCNA in solution/off DNA? You suggest that the key interaction is the one with the PIP-box and the same PIP-box allows Pol delta and many other proteins to interact with PCNA in solution/off DNA.

This is an interesting question and the phenomenon of DNA-dependent modulation of affinity to PCNA that is observed for a number of proteins including Pol δ , Pol ϵ and FEN1 (Gomes and Burgers, 2000) is puzzling. When comparing the structures of Pol ϵ and Pol δ bound to PCNA on DNA, the interaction with PCNA is in both cases mediated through the PIP-box motif that lies immediately C-terminal of the catalytic domain in the largest subunit of the two polymerases (Lancey et al., 2020, Zheng et al., 2020). For both Pol δ and Pol ϵ , these interactions are reported to be the determinants for PCNA-dependent polymerase processivity and replication fork progression rates (Johansson et al., 2004, Georgescu et al., 2014, Aria and Yeeles, 2018, Lancey et al., 2020, Zheng et al., 2020). On the other hand, the binary interaction in solution/off DNA, appears to be more complex in the case of Pol δ , which contains multiple PCNA-interacting motifs across its subunits. Particularly, a motif in the C-terminus of Pol32, the smallest subunit of *S. cerevisiae* Pol δ , was shown to be involved in the binding to PCNA in solution (Johansson et al., 2004). While this motif appears to be dispensable for processive DNA synthesis by Pol δ , the binary interaction between Pol δ and PCNA independent of DNA depends on this C-terminal interaction site in Pol32, as demonstrated by Surface Plasmon Resonance (SPR) and electrophoretic mobility shift analyses (Johansson et al., 2004). It could be the case that the additional PIP-motifs in Pol δ increase the affinity for DNA-independent interaction with PCNA. Pol ϵ has only the one PIP-box that lies C-terminal of the catalytic domain in the large subunit. The sequence of *S. cerevisiae* Dpb2 residues 210-229 resemble a PIP-box motif, but this region is rigidly folded into one long alpha helix within the non-cat domain and is unlikely to be readily accessible for an interaction with PCNA (Dua et al., 2002, Goswami et al., 2018).

Reviewer #2:

Remarks to the Author:

In this study, Roske and Yeeles report cryo-EM structures of the catalytic domain of human DNA polymerase epsilon (Pol ϵ) bound to the sliding clamp PCNA, along with various DNA substrates. Specifically, they resolved structures that represent the holoenzyme during processive elongation on a matched substrate, as well as the end-point states (referred to in the paper as 'post-insertion' and 'excision') in the pathway of transferring a mismatched substrate from the polymerase to the exonuclease site of Pol ϵ for proofreading. Additionally, two novel states (referred to as 'arrest' and 'frayed substrate') are presented, which the authors assigned as intermediates between the post-insertion and excision states.

In the cryo-EM work, constructs of Pol ϵ that included the entire POLE1 subunit along with auxiliary subunits were used. However, only the catalytic domain of POLE1 (POLE1_CAT) was resolved in the maps, suggesting that POLE1_CAT is flexibly connected to the non-catalytic domain. The structure of POLE1_CAT closely resembles that of the yeast homologue (Hogg et al, NSMB, 2014; Jain et al, PLoS one, 2014).

The general architecture of the POLE1_CAT-PCNA-DNA complexes mirrors those of other DNA polymerases bound to PCNA that have been previously published (Zheng et al, PNAS, 2020; Lancey et al, Nat Comm, 2020; Lancey et al, Nat Comm, 2021; Madru et al, Nat Comm, 2020). In these complexes, the polymerase sits atop PCNA, with DNA exiting the catalytic domain and threading through the PCNA ring. In the Pol ϵ complex, the primary interaction with PCNA involves a typical, well conserved PIP-box at the POLE1_CAT C-terminus binding to one PCNA protomer, along with additional, minor contacts between the P and Thumb domains of the polymerase and the other two PCNA protomers.

Moreover, the structure of the post-insertion state of the complex with a mismatched substrate resembles the analogous complex of human B-family Pol α -primase (Baranovskiy et al, PNAS, 2022). This similarity includes impaired stacking between the mispair and the nascent base pair and a suboptimal angle between the 3'-OH and the α -phosphate, important for catalysis. The excision state of the complex closely mirrors that of the B-family phage RB69 polymerase (Shamoo and Steitz, Cell, 1999).

The study introduces novel 'arrest' and 'frayed' states, proposed to link the post-insertion and excision states. Collectively, the authors propose a mechanism for transferring the mismatch to the polymerase and exonuclease sites, involving the expulsion of the DNA substrate from the active site by the thumb domain, fraying of two bases at the 3' end of the nascent strand, and the final 3'-end transfer to the exonuclease site. Incidentally, a "frayed primer-template" model for proof reading has been already proposed for human A-family DNA polymerase gamma, based on multiple structures of intermediates between polymerization and editing states (Park et al, NSMB, 2023; Buchel et al., Nat Comm, 2023).

While the study is technically sound, because of previous structural work revealing key aspects of DNA polymerases (particularly regarding clamp-enhanced processivity and proofreading), the authors should explain more clearly the novelty in their work.

We appreciate that the reviewer finds our work to be technically sound. To the best of our knowledge our structures are the first that show Pol ϵ bound to PCNA, which reveals a distinct mode of engagement that appears to be specialized for replication of the leading strand. Moreover, the structures on the mismatch templates provide new insights into how DNA is transferred from the polymerase to the exonuclease active site. We fully accept and acknowledge that a similar model has been proposed for Pol gamma. However, as noted, the different domain arrangement of A-family polymerases, such as Pol gamma, situates the exonuclease domain on the opposite side of the core Klenow fold when compared with B-

family polymerases and therefore requires specialised switching mechanisms for the respective polymerase families.

Major points:

- What are the structural determinants that underlie the differing functional dependencies of Pol ϵ and Pol δ on PCNA (beyond the inability of Pol ϵ -PCNA to form toolbelts with other factors)? Previous studies have demonstrated that while Pol δ is entirely dependent on PCNA for its function in DNA replication, Pol ϵ is not (see, for instance, Georgescu et al, NSMB, 2014; Lee, Hurwitz et al, JBC, 1991). The authors should further elaborate on the unique mechanisms by which PCNA enhances the processivity of Pol ϵ

As the reviewer correctly points out, yeast Pol ϵ , unlike Pol δ , was demonstrated to synthesize stretches of DNA up to ~27 nucleotides in the absence of PCNA following a single DNA binding event, which was attributed to its unique processivity (P) domain (Hogg et al., 2014).

Our view is that the primary mechanism by which PCNA enhances Pol ϵ processivity lies in its 'sliding clamp'-function to act as a tether to maintain contact between the DNA and the polymerase and prevent enzyme-substrate complex dissociation. However, we cannot exclude the existence of possibly 'unique' mechanisms of processivity enhancement. Intriguingly, while a minimal replisome (only leading strand replication) reconstituted from budding yeast proteins can achieve fast rates independent of PCNA, consistent with PCNA-independent polymerase processivity, the reconstituted human replisome displays a much stronger dependency on PCNA for efficient replication fork progression (Yeeles et al., 2017, Aria and Yeeles, 2018, Baris et al., 2022). Within the catalytic domain of Pol ϵ from both organisms, the DNA substrate is coordinated by highly conserved residues that form extensive contacts with the phosphate backbone and the grooves of the double helix. However, we notice different residue identities for the DNA-contacting sidechains in the P domain that have been identified as critical for Pol ϵ processivity (Hogg et al., 2014). Where the budding yeast homologue contacts the DNA with residues His748, Arg749 and Lys751, human Pol ϵ bears Lys732, Lys733, and His735, respectively. We also observe that unlike Arg749 in yeast Pol ϵ , Lys733 faces away from the DNA backbone in our structure, which places its primary amine too distant to form a salt bridge with the substrate DNA. Combined alanine substitution at all three positions results in loss of processivity in yeast Pol ϵ which could not be rescued by the presence of PCNA (Hogg et al., 2014), suggesting that the missing DNA-contact at Lys733 could weaken additive effects of the coordinating residues in the P domain and increase dependency on PCNA for processivity. Our structure reveals that the helix-turn in the P domain region around residue 730 is wedged between PCNA-Arg210 and -Tyr211 (**Ext. Data Fig. 2d**), keeping PCNA-Arg210 pointed toward the backbone of the template DNA in a rotamer that is different from the other two PCNA protomers. Notably, *S. cerevisiae* PCNA bears a lysine instead of an arginine residue at the equivalent position. We hypothesise that human PCNA may compensate for Pol ϵ -Lys733 in the P domain (instead of an DNA-contacting arginine residue at this position in yeast Pol ϵ) through PCNA-Arg210 which coordinates the phosphate backbone of the template strand. However, data for processivity of human Pol ϵ is currently not available, and the contribution of P-domain to processivity at the replisome remains subject to future studies, which is why we decided not to speculate on putative mechanisms beyond PCNA's function as a sliding clamp.

- I found it surprising that Pol ϵ shows interactions with multiple PCNA protomers, indicating a high-affinity interaction, which contradicts previous reports (for instance, see Chilkova et al, NAR, 2007). Without mutational or functional analysis, the significance of the novel contact points between the P and thumb domains with PCNA on Pol ϵ function remains unclear.

Here we feel that the reviewer has misinterpreted our data and the description of the data. We certainly do not think that the contacts between PCNA and the P domain and thumb contribute to a high-affinity interaction. Rather, we specifically suggested the opposite: “*The contacts formed with PCNA by Thumb and P domains are small (~270 Å² and 670 Å², respectively), mainly mediated by hydrogen bonds and not characteristic of stable protein-protein interactions, potentially allowing respective movement of the contact components.*” To quantitatively support this interpretation, we used the EMBL PDBePISA tool (Krissinel and Henrick, 2007) to measure the areas of these interfaces, as well as the theoretical solvation free energy gain upon formation of the contacts. These are calculated to be only 3.2 kcal/mol and 1.8 kcal/mol theoretical free energy gain at the interfaces at P and Thumb domains, respectively, compared to 12.5 kcal/mol free energy gain for the PIP-box interaction.

The authors should clarify why streptavidin was conjugated to one end of the DNA substrates. Was it used to prevent PCNA from sliding off the substrate and to enhance its affinity for Pol ε?

Based on previous reports, we anticipated a low-affinity interaction between Pol ε and PCNA and included a 5'-biotin modification at the blunt end of the duplex region of the 23/38 nucleotide nascent strand/template strand DNA substrate. As depicted in **Ext. Data Fig. 1b**, this allows the conjugation of streptavidin to the blunt end, which was intended to aid in the stability of the complex once it was formed, i.e. to prevent PCNA from sliding off the DNA during the size-exclusion chromatography step. However, we found that streptavidin was dispensable in the optimized final conditions of complex assembly. No streptavidin was added in the samples on matched DNA that was vitrified in the presence of CHAPSO detergent. Similarly, no streptavidin was added in the sample preparation on the mismatched DNA substrate which does not contain a biotin modification. We have added clarifying statements in the manuscript (see **Methods** and figure legend of **Ext. Data Fig. 1b**).

- The observed flexibility between POLE1_CAT, the POLE1 dead domain, and ancillary subunits in these structures contrasts with a previous structural report on the yeast homolog, which depicted a rigid complex (Yuan, Nat Comm, 2020). The authors should discuss this discrepancy and evaluate whether the interactions of POLE1_CAT with PCNA could also occur in a rigid Pol ε complex.

We thank the author for raising this point. We think it is worth mentioning that in almost all published cryo-EM data sets of yeast and human Pol ε lacking PCNA, both in isolation and bound to CMG, the majority of 3D classes display significant flexibility between the POLE1 cat and non-cat domains. Therefore, the data we present is consistent with the bulk of the published literature.

To evaluate whether the interaction with PCNA can occur in the rigid Pol ε complex, we have performed molecular docking of our Pol ε cat-PCNA structure and with said structure of *S. cerevisiae* Pol ε which includes the ancillary domains. The PCNA interaction of the ternary complex can be accommodated in this proposed ‘linear’ configuration of the two Pol ε lobes, without clashes. Despite the proximity of the ancillary domains to the Thumb domain of Pol ε cat, the large conformational changes that we describe for the Thumb can also be accommodated in this ‘linear’ configuration. In addition to the ‘linear’ configuration of Pol ε (that we also previously observed for human Pol ε bound to CMG (Jones et al., 2021)), we have observed the catalytic domain of yeast Pol ε in a relatively stable ‘folded’ configuration where it is positioned directly under the C-tier of MCM (Jenkyn-Bedford et al., 2021). Similar to the ‘linear’ configuration, it is possible to dock our model of Pol ε cat-PCNA onto the ‘folded’ configuration without clashes. Currently it is unclear how these distinct conformations of Pol ε cat relate to active leading-strand replication. Therefore, in the revised manuscript we have included a new figure (**Ext. Data Fig. 6**) showing that PCNA engagement is compatible with

both arrangements. Furthermore, the reported interaction with the Ctf18-1-8 module of PCNA clamp loader Ctf18-RFC is also compatible with our model for the complex of Pol ϵ cat, PCNA and DNA (**Ext. Data Fig. 6d**).

- The resolution of the 'frayed substrate' state map at 4.14Å poses challenges for side chain modeling. The map lacks density for the side chains of residues K732, K733, and R728 in Pol ϵ , yet the authors suggest these residues help tightly bind the DNA to the polymerase. Additionally, the weak density for the 3'-end of the nascent strand and its apparent looseness raise questions about the interactions responsible for melting from the template strand.

We agree with the reviewer that at resolutions below 4 Å, density for many amino acid side chain residues is lacking. Nevertheless, the trace of the protein backbone and the locations of C β remain clear, and together with the density for the DNA phosphate backbone in the described region, we are confident that the DNA is closely coordinated at the P domain in the Frayed state. Although the precise rotamers of the noted amino acid side chain residues cannot be clearly determined from the structure, the close contact between P domain and the nascent DNA strand (<5 Å distance between the main chain amide and the phosphate backbone) remains clearly distinguished from the other states. We reworded the section describing this contact to: *"In this configuration, the coordination of DNA at the P domain is more extensive than we observe in structures with matched DNA, with the backbone of the nascent strand held tightly against the positively charged surface of the P domain formed by residues Arg728, Lys733 and His735 (Fig. 3e)."*

Indeed, the 3'-end of the nascent strand is only loosely positioned at the entry site of the exonuclease domain in the Frayed Substrate state, which is indicated by the weak density. The same applies to the single-stranded segment of the template strand in the Frayed and the Excision states. Based on these observations, our model does not assume interactions with the single-stranded portions of the DNA substrate in the Frayed state to be responsible for strand melting. Rather, we propose that it is the handover of the double-stranded portion of the DNA between P domain and Thumb that first allows capture of the mismatched base near the entry site of the exonuclease domain and subsequently drive the melting of additional bases to channel the erroneous base into the exonuclease cavity for excision. We have adjusted the results section describing the Frayed Substrate state as well as the description of our model for activity switching in the discussion to convey the proposed mechanism more clearly.

- The authors should clarify the structural determinants that prevent the fingers domain from closing in the post-insertion state, despite the presence of a complementary nucleotide in the active site.

The presence of a complementary nucleotide induces Finger closing at a matched, but not the mismatched substrate, which suggests that the Finger closing conformational change is dependent on correct geometry of the DNA substrate and thus sensitive toward incorporated mismatches. As noted, however, the structures of the Pol ϵ replication conformers with opened Fingers at matched and mismatched DNA substrates are highly similar (rmsd = 0.87 Å for 1116 C α atoms) and no obvious structural features pose steric barriers to fingers closing. We do notice subtle differences between the two structures, which are described in the results section under '**Pol ϵ ternary complex with a T-C mismatch in the postinsertion site.**' to the extent that the resolution of 3.3 Å allows. These include a) a small distortion of the T-C mismatched nascent strand in the postinsertion site, which result in b) water-mediated rather than hydrogen-bonded base pairing of the mismatched pair, c) imperfect base stacking with the incoming nucleotide and d) increased flexibility in the region of Pol ϵ that surrounds the active site and the mismatched penultimate base pair, which may cumulatively disfavour Fingers closing.

- For enhanced clarity, it would be beneficial if the authors could produce morphing movies that show transitions between the four proofreading states, highlighting the evolution of key protein-DNA interactions

During the analysis of our data, we too thought that morphing movies would benefit the illustration of the conformational changes between the states of activity switching. However, we found the interpolation between the states, especially in the transition from the Arrest to the Frayed state, could be highly misleading and not representative of what may actually occur during the conformational transitions and the DNA handover. Consequently, we decided against including the morphing movies in the original submission. Our view on this has not changed and therefore we have not included morphing movies with the revised submission.

Minor points:

- The authors say that “Pol δ binds PCNA via a highly conserved PCNA-interacting peptide (PIP-box) motif in the large polymerase subunit p125 (or POLD1)”. However, the PIP-box is not highly conserved and is atypical (Zheng et al, PNAS, 2020; Lancey et al, Nat Comm, 2020). In addition, other interactions contribute in binding the Pol to PCNA.

We thank the reviewer for pointing out the imprecision in our wording. We intended to describe the PIP-box that is conserved within Pol δ across different species and have amended this accordingly.

- What are the axes of rotation shown in Fig. 2b and Fig. 5b?

For clarity, we have added information in the figure legend for 2b explaining which other panels the rotation symbols are relative to. Similarly, in the figure legend for 5b, we added ‘left’ to the current description to clarify that the rotation symbol indicates the view relative to the left view in panel 5a.

Reviewer #3:

Remarks to the Author:

This manuscript presents cryo-EM structures of human DNA polymerase epsilon in the process of transitioning a T•C mismatch from the polymerase active site to the exonuclease active site to remove the incorrect terminal nucleotide from single-stranded DNA. In my opinion, the study is beautifully performed and described, and it is worthy of publication in NSMB. The new structures imply that proofreading by this enzyme can occur without enzyme dissociation, a point that will be of great interest to experts interested in the mechanisms that determine accurate replication. The study is highly relevant to the emerging data published by others suggesting that proofreading by Pol epsilon, which is hypothesized to primarily replicate the leading DNA strand during replication, differs somewhat from DNA polymerase delta. (This could possibly be discussed in greater detail, either here or at some point in the future.) In addition, the study should be of great interest to scientists who study the origins of human diseases, and for several reasons, to evolutionary biologists. I congratulate these two authors on a fine study. The observations made here will form the baseline for much future work.

On what basis was a T•C mismatch chosen for this study? Are there any issues worth considering given that it is but one mismatch among billions that are potentially relevant to replication of the human nuclear genome?

We decided to use a T-C mismatch because we expected it to be sterically undemanding compared to, for example, purine-purine mismatches and thus able to be accommodated in the post-insertion site, as observed in other polymerases in replication conformers (Baranovskiy et al., 2022, Johnson and Beese, 2004). Further, to characterise mismatch sensing and activity switching, we decided against using a DNA substrate with an artificially pre-formed frayed 3'-end (i.e. more than one mismatched base pair) such as used in (Gouge et al., 2012). Interestingly, while proofreading-proficient Pol ϵ is among the highest fidelity polymerases, the base-substitution fidelity of Pol ϵ *exo-* was observed lower when comparing with other exonuclease-deficient replicative DNA polymerases, with elevated rates of pyrimidine-pyrimidine mispairs, such as a T-C (Shcherbakova et al., 2003), making this mismatch a relevant substrate to study in the context of mismatch excision.

- ARIA, V. & YEELES, J. T. P. 2018. Mechanism of Bidirectional Leading-Strand Synthesis Establishment at Eukaryotic DNA Replication Origins. *Mol Cell*, 73, 199-211 e10.
- BARANOVSKIY, A. G., BABAYEVA, N. D., LISOVA, A. E., MORSTADT, L. M. & TAHIROV, T. H. 2022. Structural and functional insight into mismatch extension by human DNA polymerase alpha. *Proc Natl Acad Sci U S A*, 119, e2111744119.
- BARIS, Y., TAYLOR, M. R. G., ARIA, V. & YEELES, J. T. P. 2022. Fast and efficient DNA replication with purified human proteins. *Nature*, 606, 204-210.
- DUA, R., LEVY, D. L., LI, C. M., SNOW, P. M. & CAMPBELL, J. L. 2002. In vivo reconstitution of *Saccharomyces cerevisiae* DNA polymerase epsilon in insect cells. Purification and characterization. *J Biol Chem*, 277, 7889-96.
- GEORGESCU, R. E., LANGSTON, L., YAO, N. Y., YURIEVA, O., ZHANG, D., FINKELSTEIN, J., AGARWAL, T. & O'DONNELL, M. E. 2014. Mechanism of asymmetric polymerase assembly at the eukaryotic replication fork. *Nat Struct Mol Biol*, 21, 664-70.
- GOMES, X. V. & BURGERS, P. M. 2000. Two modes of FEN1 binding to PCNA regulated by DNA. *EMBO J*, 19, 3811-21.
- GOSWAMI, P., ABID ALI, F., DOUGLAS, M. E., LOCKE, J., PURKISS, A., JANSKA, A., EICKHOFF, P., EARLY, A., NANS, A., CHEUNG, A. M. C., DIFFLEY, J. F. X. & COSTA, A. 2018. Structure of DNA-CMG-Pol epsilon elucidates the roles of the non-catalytic polymerase modules in the eukaryotic replisome. *Nat Commun*, 9, 5061.
- GOUGE, J., RALEC, C., HENNEKE, G. & DELARUE, M. 2012. Molecular recognition of canonical and deaminated bases by P. abyssi family B DNA polymerase. *J Mol Biol*, 423, 315-36.
- HOGG, M., OSTERMAN, P., BYLUND, G. O., GANAI, R. A., LUNDSTROM, E. B., SAUER-ERIKSSON, A. E. & JOHANSSON, E. 2014. Structural basis for processive DNA synthesis by yeast DNA polymerase varepsilon. *Nat Struct Mol Biol*, 21, 49-55.
- JENKYN-BEDFORD, M., JONES, M. L., BARIS, Y., LABIB, K. P. M., CANNONE, G., YEELES, J. T. P. & DEEGAN, T. D. 2021. A conserved mechanism for regulating replisome disassembly in eukaryotes. *Nature*, 600, 743-747.
- JOHANSSON, E., GARG, P. & BURGERS, P. M. 2004. The Pol32 subunit of DNA polymerase delta contains separable domains for processive replication and proliferating cell nuclear antigen (PCNA) binding. *J Biol Chem*, 279, 1907-15.

- JOHNSON, S. J. & BEESE, L. S. 2004. Structures of mismatch replication errors observed in a DNA polymerase. *Cell*, 116, 803-16.
- JONES, M. L., BARIS, Y., TAYLOR, M. R. G. & YEELES, J. T. P. 2021. Structure of a human replisome shows the organisation and interactions of a DNA replication machine. *EMBO J*, 40, e108819.
- KRISSINEL, E. & HENRICK, K. 2007. Inference of macromolecular assemblies from crystalline state. *J Mol Biol*, 372, 774-97.
- LANCEY, C., TEHSEEN, M., RADUCANU, V. S., RASHID, F., MERINO, N., RAGAN, T. J., SAVVA, C. G., ZAHER, M. S., SHIRBINI, A., BLANCO, F. J., HAMDAN, S. M. & DE BIASIO, A. 2020. Structure of the processive human Pol delta holoenzyme. *Nat Commun*, 11, 1109.
- SHCHERBAKOVA, P. V., PAVLOV, Y. I., CHILKOVA, O., ROGOZIN, I. B., JOHANSSON, E. & KUNKEL, T. A. 2003. Unique error signature of the four-subunit yeast DNA polymerase epsilon. *J Biol Chem*, 278, 43770-80.
- YEELES, J. T. P., JANSKA, A., EARLY, A. & DIFFLEY, J. F. X. 2017. How the Eukaryotic Replisome Achieves Rapid and Efficient DNA Replication. *Mol Cell*, 65, 105-116.
- ZHENG, F., GEORGESCU, R. E., LI, H. & O'DONNELL, M. E. 2020. Structure of eukaryotic DNA polymerase delta bound to the PCNA clamp while encircling DNA. *Proc Natl Acad Sci U S A*, 117, 30344-30353.

Decision Letter, first revision:

Message: Our ref: NSMB-A49016A

16th May 2024

Dear Dr. Yeeles,

Thank you for submitting your revised manuscript "Structural basis for processive daughter strand synthesis and proofreading by the human leading-strand polymerase Pol ϵ " (NSMB-A49016A). It has now been seen by the original referees and their comments are below. The reviewers find that the paper has improved in revision, and therefore we are happy to accept it in principle in Nature Structural & Molecular Biology, pending minor revisions to satisfy the referees' final requests and to comply with our editorial and formatting guidelines.

We are now performing detailed checks on your paper and will send you a checklist detailing our editorial and formatting requirements in about two weeks. Please do not upload the final materials and make any revisions until you receive this additional information from us.

To facilitate our work at this stage, it is important that we have a copy of the main text as a word file. If you could please send along a word version of this file as soon as possible, we would greatly appreciate it; please make sure to copy the NSMB account (cc'ed above).

Sincerely,

Dimitris Typas
Senior Editor
Nature Structural & Molecular Biology
ORCID: 0000-0002-8737-1319

Reviewer #1 (Remarks to the Author):

I am happy with the answers and congratulate the authors on a very good paper.

Reviewer #2 (Remarks to the Author):

The authors have addressed all my concerns satisfactorily, and I recommend the paper for publication. I congratulate the authors for their work.

Reviewer #3 (Remarks to the Author):

My previous comments give my opinion, which stands as previously stated. It will of course eventually be necessary confirm the statements and interpretations made here using other mismatches, but this study is sufficient for publication as is.

Final Decision Letter:

Message: 11th Jul 2024

Dear Dr. Yeeles,

We are now happy to accept your revised paper "Structural basis for processive daughter strand synthesis and proofreading by the human leading-strand DNA polymerase Pol ϵ " for publication as an Article in Nature Structural & Molecular Biology.

As soon as your article is published, you can generate your shareable link by entering the DOI of your article here: <http://authors.springernature.com/share>. Corresponding authors

will also receive an automated email with the shareable link

Your paper will be published online soon after we receive proof corrections and will appear in print in the next available issue. You can find out your date of online publication by contacting the production team shortly after sending your proof corrections.

Please note that *Nature Structural & Molecular Biology* is a Transformative Journal (TJ). Authors may publish their research with us through the traditional subscription access route or make their paper immediately open access through payment of an article-processing charge (APC). Authors will not be required to make a final decision about access to their article until it has been accepted. Find out more about Transformative Journals

Authors may need to take specific actions to achieve compliance with funder and

institutional open access mandates. If your research is supported by a funder that requires immediate open access (e.g. according to Plan S principles) then you should select the gold OA route, and we will direct you to the compliant route where possible. For authors selecting the subscription publication route, the journal's standard licensing terms will need to be accepted, including self-archiving policies. Those licensing terms will supersede any other terms that the author or any third party may assert apply to any version of the manuscript.

Sincerely,

Dimitris Typas
Senior Editor
Nature Structural & Molecular Biology
ORCID: 0000-0002-8737-1319